# Exploring the role of the outer subventricular zone during cortical folding through a physics-based model

Mohammad Saeed Zarzor[1], Ingmar Blumcke[2], Silvia Budday[1]*

[1]Friedrich-Alexander-Universität Erlangen-Nürnberg, Institute of Applied Mechanics, Erlangen, Germany; [2]University Hospitals Erlangen, Institute of Neuropathology, Erlangen, Germany

**Abstract** The human brain has a highly complex structure both on the microscopic and on the macroscopic scales. Increasing evidence has suggested the role of mechanical forces for cortical folding – a classical hallmark of the human brain. However, the link between cellular processes at the microscale and mechanical forces at the macroscale remains insufficiently understood. Recent findings suggest that an additional proliferating zone, the outer subventricular zone (OSVZ), is decisive for the particular size and complexity of the human cortex. To better understand how the OSVZ affects cortical folding, we establish a multifield computational model that couples cell proliferation in different zones and migration at the cell scale with growth and cortical folding at the organ scale by combining an advection-diffusion model with the theory of finite growth. We validate our model based on data from histologically stained sections of the human fetal brain and predict 3D pattern formation. Finally, we address open questions regarding the role of the OSVZ for the formation of cortical folds. The presented framework not only improves our understanding of human brain development, but could eventually help diagnose and treat neuronal disorders arising from disruptions in cellular development and associated malformations of cortical development.

*For correspondence: silvia.budday@fau.de

Competing interest: The authors declare that no competing interests exist.

## Editor's evaluation

Through theoretical analysis, the authors argue that the proliferation of neurons in the outer subventricular zone, which is specific to humans, decreases the distance between neighboring sulci in the cerebral cortex and increases cell density in the ventricular zone. Though the exact mechanisms remain to be further elucidated, the compelling data and approach represent a valuable foundation for the study of cortical folding from the underpinning cellular level as well as the coupling role of mechanics and cellular biology. This study will be of particular interest to the large community of scientists studying the mechanisms of brain development and disorder and even possibly beyond.

## Introduction

The brain is one of the most fascinating organs in the human body. Its complex structure on both micro- and macroscopic scales closely correlates with the unique cognitive abilities of humans. Cortical folding is one of the most important features of the human brain. Still, compared to other mammals, the human brain is neither the largest nor the most folded brain. However, relative to its size, it has the largest number of cortical neurons that connect with billions of neuronal synapses (*Herculano-Houzel, 2009*). This fact attracted the attention of neuroscientists over the past few years to explore the source of these cells and how they develop in the early stages of brain development.

The number of brain cells is determined in utero through the proliferation process. Previous studies on different lissencephalic species, such as mice, have shown that cell division in the brain is confined to a small region near the cerebral ventricles (*Hansen et al., 2010*). However, in gyrencephalic species, this seems to be different. Recent findings show that the human brain, for example, is characterized by two proliferation zones with two different types of progenitor cells. Both zones produce neurons that later migrate towards the outer brain surface and form the cortex *Lui et al., 2011*; *Pebworth et al., 2021*.

In rodents, progenitor cells around the ventricular zone (VZ) generate intermediate progenitor cells as their daughters, which accumulate above the VZ and form a new layer called the subventricular zone (*Noctor et al., 2002*). In humans, there is an additional outer layer of the subventricular zone, often referred to as outer subventricular zone (OSVZ) (*Hansen et al., 2010*; *Lui et al., 2011*; *Noctor et al., 2007*). This zone was first discovered in the monkey brain by Colette Dehay and her colleagues (*Smart et al., 2002*), and confirmed in the human brain by several following studies (*Huttner and Kosodo, 2005*). The OSVZ seems to play a significant role in the proliferation process and affects the size and complexity of the human cortex. The evidence for this allegation is the wave of cortical neurogenesis that coincides with the cell division in the OSVZ (*Lukaszewicz et al., 2005*). At the macroscopic scale, the high proliferation in the OSVZ coincides with a significant tangential expansion of the cortical layers. The latter is an essential factor for the formation of cortical folds (*Reillo et al., 2011*). Still, it remains unknown, how exactly this proliferation process in the OSVZ affects gyrification of the forming cortex. Different approaches have been used to understand the relation between cellular mechanisms at the microscopic scale and cortical development at the macroscopic scale. Genetic analyses and experimental studies using cell culture models and brain organoids have given first valuable insights concerning the source of cells and their behavior (*Hansen et al., 2010*). Here, we intend to complement these studies by using a numerical approach to bridge the scales from the behavior of different progenitor cell types at the cell scale to the emergence of cortical folds at the tissue or organ scale.

From a mechanics point of view, forces that are generated due to cellular processes may act as a link to understand the underlying mechanisms behind cortical folding (*Budday et al., 2015b*). Many previous studies tried to explain normal and abnormal cortical folding either from a purely biological or mechanical perspective (*Tallinen et al., 2014*; *Razavi et al., 2015*). However, it will not be possible to capture the folding mechanism without considering both perspectives at the same time (*de Rooij and Kuhl, 2018*; *Zarzor et al., 2021*; *Wang et al., 2022*). In other words, to fully understand the physiological and pathological mechanisms underlying cortical folding in the developing human brain, we need to study the coupling between cellular processes and mechanical forces – to eventually assess how disruption of cellular processes affect the folding pattern and lead to malformations of cortical development (*Guerrini et al., 2008*; *Blumcke et al., 2021*; *Llinares-Benadero and Borrell, 2019*).

To fill this knowledge gap, we establish a two-field computational model that accounts for both proliferating zones in the human brain, the VZ and OSVZ. The first field in the model describes the growth and deformation of brain tissue based on the theory of finite growth (*Rodriguez et al., 1994*; *Göktepe et al., 2010*). The second field describes the cellular processes occurring during human brain development, where we use an advection-diffusion equation to mimic the proliferation and migration in the subcortex and neuronal connectivity in the cortex (*de Rooij and Kuhl, 2018*; *Zarzor et al., 2021*). We add two source terms to consider the division in different zones. We validate our model through a comparison of the simulation results with histologically stained sections of the human fetal brain and address unresolved questions regarding the role of the OSVZ for cortical folding.

## Cellular processes during brain development

The central cellular unit that plays a critical role in essential processes of brain development is a type of progenitor cells called radial glial cells. In the early stage of brain development, when neurogenesis begins, neuroepithelial cells transform into radial glial cells (*Noctor et al., 2007*). Around gestational week 5, these cells locate near cerebral ventricles, in the VZ, where they undergo interkinetic nuclear migration. The associated symmetric division behavior leads to a significant increase in the number of radial glial cells and results in both increased thickness and surface area of the VZ (*Blows, 2003*; *Fish et al., 2008*; *Bystron et al., 2008*). Subsequently, the cells switch to an asymmetric division behavior and generate intermediate progenitor cells (*Noctor et al., 2004*). The latter migrate to the

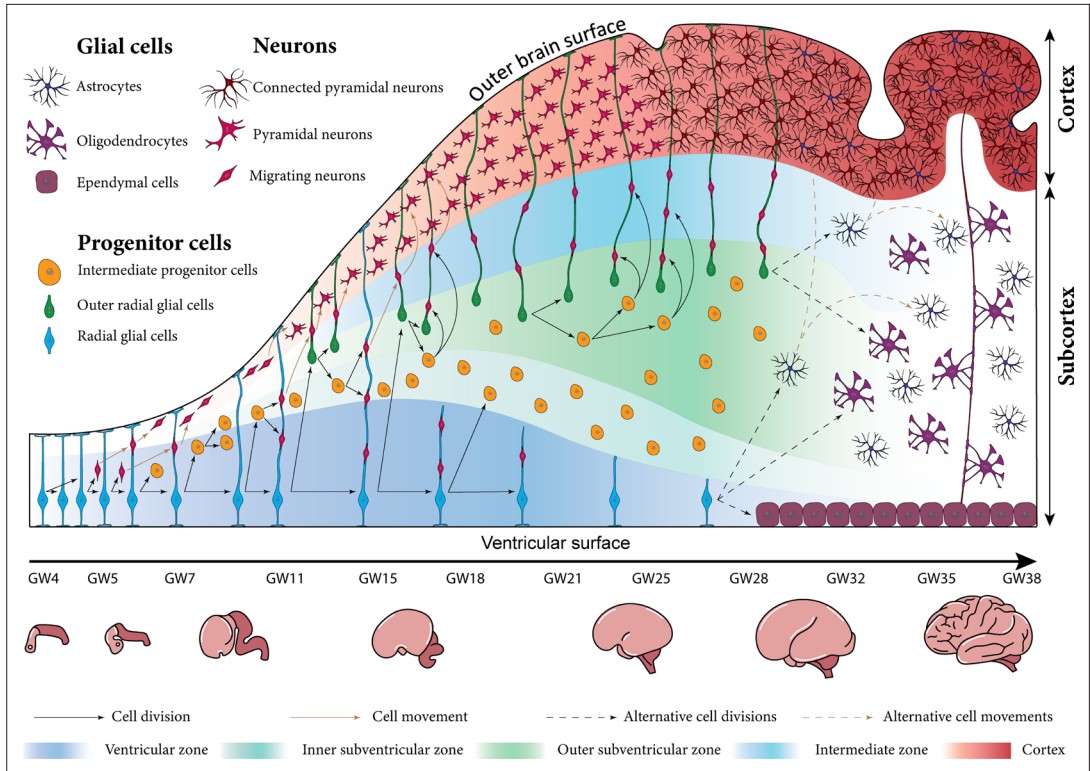

**Figure 1.** Schematic illustration of human brain development between gestational weeks (GW) 4 and 38 at the cellular scale (top) and the organ scale (bottom). In the early stage of development, the repetitive division of radial glial cells in the ventricular zone (VZ) significantly increases the total number of brain cells. The newly born intermediate progenitor cells accumulate above the VZ and form a new layer called the inner subventricular zone. The outer radial glial cells (ORGCs) that are produced around gestational week 11 form a new layer called the outer subventricular zone (OSVZ). The neurons generated from progenitor cells migrate along radial glial cell fibers towards the cortex. Around gestational week 28, the migration process is almost finished, and the radial glial cells switch to produce different types of glial cells like astrocytes and oligodendrocytes.

subventricular zone, where they proliferate and produce neurons, as illustrated in *Figure 1* (*Noctor et al., 2007*; *Pebworth et al., 2021*). Ultimately, the majority of cortical neurons are produced by intermediate progenitor cells (*Lui et al., 2011*; *Libé-Philippot and Vanderhaeghen, 2021*).

According to the radial unit hypothesis proposed by Pasko Rakic over 30 years ago, the radial glial cell fibers organize the migration process, which starts around gestational week 6 (*Nonaka-Kinoshita et al., 2013*). He postulated that these fibers form a scaffold to guide neurons during their migration from the proliferating zones to their final destination in the cortex, which forms the outer brain surface (*Rakic, 1988*; *Lui et al., 2011*). In gyrencephalic species, those fibers have a characteristic fan-like distribution (*Borrell and Götz, 2014*; *Nonaka-Kinoshita et al., 2013*). However, it is still under debate whether this unique distribution is the cause or rather the result of cortical folding. Our recent computational analyses support the latter, i.e., that it is the result of cortical folding (*Zarzor et al., 2021*). The migration process synchronizes with a radial expansion of all brain layers. Still, the VZ does not expand remarkably as the intermediate progenitor cells move outwards to the subventricular zone. The migrated neurons finally organize themselves in the six-layered cortex in an inside-out sequence, where the early-born neurons occupy the inner layers (*Gilmore and Herrup, 1997*). Until gestational week 23, the outer brain surface is still smooth, although the bottom four layers of the cortex are already filled with neurons (*Shinmyo et al., 2017*). The first folds appear between gestational weeks 20 and 28, as the cortical layer significantly expands tangentially (*Budday et al., 2015b*; *Habas et al., 2012*). Importantly, at around gestational week 25, the cortical neuronal connectivity emerges and comes along with the horizontal elongation of neuronal dendrites (*Takahashi et al., 2012*).

While the processes summarized above are common among mammals, the human brain has some specific features that play a significant role in increasing the number of cortical neurons, and enhancing

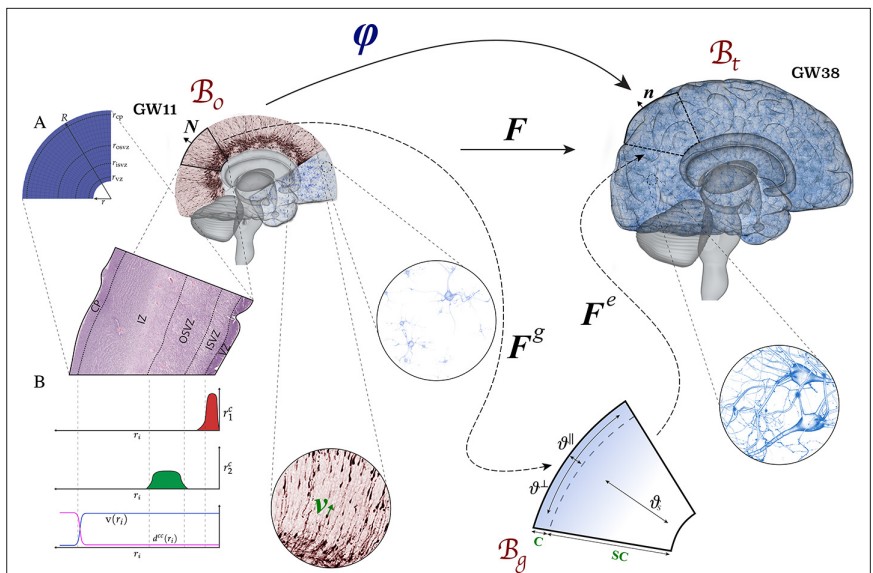

**Figure 2.** Kinematics of the multifield brain growth model. The reference configuration $\mathcal{B}_0$ represents the initial state of the brain at gestational week (GW) 11. The spatial configuration $\mathcal{B}_t$ represents the state of the brain at any time $t$ during development. The stress-free (intermediate) growth configuration $\mathcal{B}_g$ is inserted between reference and spatial configurations. (**A**) Simulation domain representing a part of the human brain's frontal lobe. (**B**) Distribution of model parameters $(r_1^c, r_2^c, v, and\ d^{cc})$ along the brain's radial direction $r_i$ from the ventricular surface to the outer cortical surface.

the complexity of cortical folds (**Libé-Philippot and Vanderhaeghen, 2021**). At the beginning of the second trimester, around gestational week 11, the original radial glial cells switch from producing intermediate progenitor cells to producing a special kind of cells that is found in all gyrencephalic species but is enriched in the human brain. The newly generated cells are similar to the radial glial cells in terms of shape and function, but unlike the original radial glial cells, they migrate to the outer layer of the subventricular zone, referred to as OSVZ, after they are born. Therefore, they are called outer radial glial cells (ORGCs) (**Lui et al., 2011**; **Fietz et al., 2010**; **Hansen et al., 2010**; **Reillo et al., 2011**; **Nonaka-Kinoshita et al., 2013**). We would like to note that some literature refers to this type of cells as basal radial glial cells. While the original radial glial cells have a bipolar morphology with two processes – one extending to the cerebral ventricle and one to the outer cortical surface – the ORGCs have a distinct unipolar structure with only a single process extending to the outer cortical surface (**Hansen et al., 2010**; **Betizeau et al., 2013**; **Reillo et al., 2011**; **Nonaka-Kinoshita et al., 2013**).

The OSVZ shows a significantly more pronounced radial expansion compared to the inner subventricular zone and VZ between gestational weeks 11.5 and 32. The immediate reason causing this difference is the characteristic division behavior of ORGCs: they translocate rapidly in radial direction before they divide, which scientists have referred to as 'mitotic small translocation (MST)' (**Fietz et al., 2010**). Importantly, the MST behavior pushes the boundary of the OSVZ outward, which increases its capacity to produce new neurons. The intermediate progenitor cells have enough space to undergo multiple rounds of division before producing neurons, which increases the overall number of generated neurons (**Kriegstein et al., 2006**; **Lui et al., 2011**).

The ORGCs, like radial glial cells, play an important role in the proliferation process: they divide symmetrically and asymmetrically to produce further ORGCs and intermediate progenitor cells (**Libé-Philippot and Vanderhaeghen, 2021**). Intermediate progenitor cells divide to generate a pair of neurons (**Lui et al., 2011**). According to previous studies, 40% of produced neurons are generated by ORGCs at gestational week 13, but this ratio increases to 60% by gestational week 14, and exceeds 75% by gestational week 15.5. Then, after gestational week 17, the ORGCs become the only source of cortical neurons in the upper cortical layers (**Hansen et al., 2010**). Besides their role in increasing the number of neurons, the ORGCs generate additional scaffolds that elongate to the outer brain surface and serve as paths for neuronal migration (**Llinares-Benadero and Borrell, 2019**; **Nonaka-Kinoshita et al., 2013**). Compared to other mammals, the neurogenesis of the human cortex can

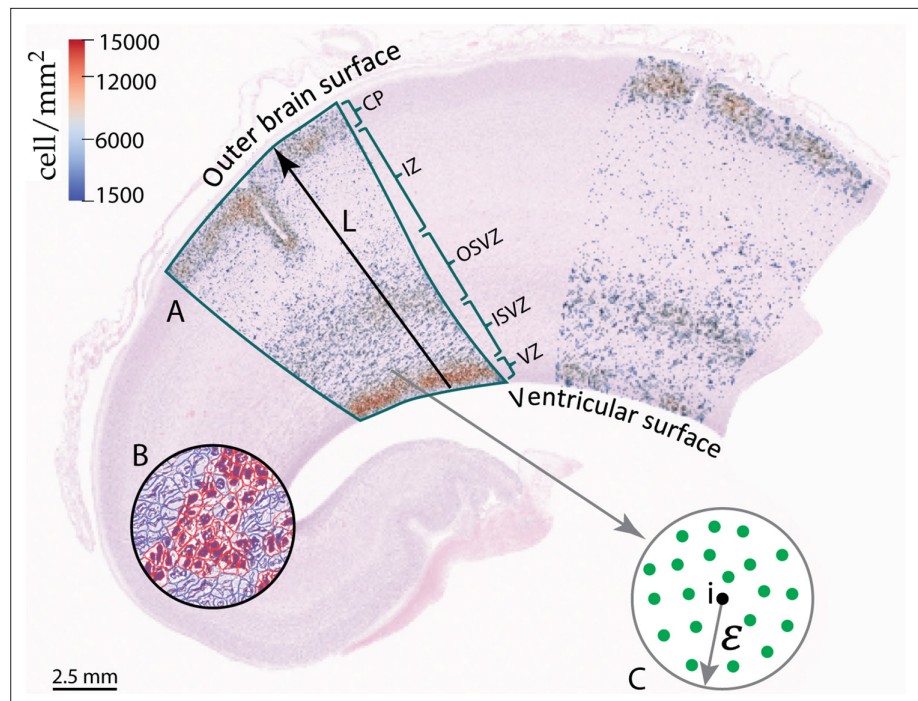

**Figure 3.** Part of the frontal lobe of a histologically stained section of the human fetal brain at gestational week 17. (**A**) Annotated area with final cell density distribution. (**B**) Example of cell detection by using *Qupath*. Red cells depict neurons and blue cells glial cells. (**C**) Procedure to determine the cell density.

thus be divided into two main stages. The first stage is characterized by migration along a continuous scaffold consisting of radial glial cell fibers, which run from the ventricular surface to the outer cortical layer around gestational week 15. During the second stage, the migration path switches to a discontinuous form. After gestational week 17, the radial glial cell fibers run from the ventricular surface to the inner subventricular zone, while ORGC fibers run from the OSVZ to the outer brain surface, as indicated in *Figure 1* (*Nowakowski et al., 2016*). Consequently, migrating neurons follow a sinuous path through numerous radial fibers before they reach its final location in the cortex (*Lui et al., 2011*). The additional scaffolds formed by ORGC fibers are not only important for neuronal migration, but also for the tangential expansion of the cortex. Previous studies show that a reduced number of ORGCs lead to reduced tangential expansion. In these cases, the cortex is less folded or even lissencephalic (*Poluch and Juliano, 2015*). In contrast, increasing the number of ORGCs leads to more excessive folding (*Florio et al., 2017*; *Borrell, 2018*). Still, it remains unknown whether these effects are a result of the specific proliferation behavior of ORGCs or the associated scaffold of ORGC fibers. What is known, though, is that the existence of ORGCs is a necessary but not sufficient condition for cortical folding (*Llinares-Benadero and Borrell, 2019*).

Around gestational week 28, the migrating neurons occupy the first (top) cortical layer, while the migration and proliferation processes come to an end. After finishing their role during the neurogenesis stage, radial glial cells and ORGCs switch to produce different types of glial cells, e.g., astrocytes and oligodendrocytes (*Schmechel and Rakic, 1979*). Also, radial glial cells may later convert into ependymal cells that locate around the cerebral ventricles. The oligodendrocytes form myelin sheaths around neuronal axons, wherefore the subcortical layer gains its characteristic white color.

## Model

To numerically study the effect of the VZ and the OSVZ on the resulting folding pattern, we simulate human brain development by using the finite element method. The influence of various factors on the emergence of cortical folds can be best shown on a simple two-dimensional (2D) quarter-circular geometry (*Darayi et al., 2022*), as illustrated in *Figure 2A*. In addition, we also investigate the folding evolution on a simplified half-sphere three-dimensional (3D) geometry. In the following, we introduce

the main equations describing the coupling between cellular mechanisms in different proliferating zones and cortical folding, which we solve numerically.

## Kinematics

To mathematically describe brain growth, we use the theory of nonlinear continuum mechanics supplemented by the theory of finite growth (**Rodriguez et al., 1994**; **Göktepe et al., 2010**). The initial state of the brain at an early stage of development, around gestational week 11, is represented by the reference configuration $\mathcal{B}_0$. The state of the brain at time $t$ later during development is represented by the spatial configuration $\mathcal{B}_t$. The deformation map $\boldsymbol{x} = \boldsymbol{\varphi}(\boldsymbol{X}, t)$ maps a reference point $\boldsymbol{X} \in \mathcal{B}_0 \subset \mathbb{R}^3$ to its new position $\boldsymbol{x} \in \mathcal{B}_t \subset \mathbb{R}^3$ at a specific time $t$, as illustrated in *Figure 2*. The derivative of the deformation map with respect to reference point position vector is called deformation gradient $\boldsymbol{F} = \nabla_{\boldsymbol{X}} \boldsymbol{\varphi}$. The local volume change of a volume element is described by the Jacobian $J = \det \boldsymbol{F}$.

Following the theory of finite growth (**Rodriguez et al., 1994**; **Göktepe et al., 2010**), we introduce a stress-free configuration between the reference and spatial configuration, the growth configuration $\mathcal{B}_g$. Accordingly, the deformation gradient is multiplicatively decomposed into an elastic deformation tensor $\boldsymbol{F}^e$ and a growth tensor $\boldsymbol{F}^g$, such that,

$$\boldsymbol{F} = \boldsymbol{F}^e \cdot \boldsymbol{F}^g. \tag{1}$$

The elastic deformation tensor describes the purely elastic deformation of the brain under the effect of external forces or forces generated internally to preserve tissue continuity. On the other hand, the growth tensor controls the amount and directions of unconstrained expansion. We note that the elastic deformation tensor is reversible, while the growth tensor is not.

To not only predict brain growth but also its relation to cellular processes during brain development, we introduce the spatial cell density $c(\boldsymbol{x}, t)$ as an additional scalar independent field that depends on the spatial point position and time. It represents the number of neurons per unit area (**de Rooij and Kuhl, 2018**) in 2D and per unit volume in 3D. The corresponding balance equation describes cell division – resulting in newborn cells – through appropriate source terms and cell migration – the directed movement of neurons – through appropriate flux terms.

For the two unknown fields, the deformation and the cell density, we introduce not only balance but also constitutive equations in the following that then allow us to compute their evolution in space and time through numerical simulations. We explain how we mathematically describe the mechanical (growth) problem and cellular processes as well as how those are linked to capture feedback mechanisms between cellular processes, mechanics, and growth.

## Mechanical problem

To govern the mechanical problem, we use the balance of linear momentum given in the spatial configuration $\mathcal{B}_t$,

$$\nabla_{\boldsymbol{x}} \cdot \boldsymbol{\sigma} = \boldsymbol{0} \qquad \text{with} \qquad \boldsymbol{\sigma} = \boldsymbol{\sigma}(\boldsymbol{F}^e), \tag{2}$$

where $\nabla_{\boldsymbol{x}}$ is the spatial gradient operator and $\boldsymbol{\sigma}$ is the Cauchy stress tensor formulated in terms of the elastic deformation tensor, as only the elastic deformation induces stresses. The Cauchy stress describes the 3D stress state in the spatial (grown and deformed) configuration and is computed by deriving the strain energy function $\psi_g$ with respect to elastic deformation tensor,

$$\boldsymbol{\sigma}(\boldsymbol{F}^e) = \frac{1}{J^e} \frac{\partial \psi_g(\boldsymbol{F}^e)}{\partial \boldsymbol{F}^e} \cdot \boldsymbol{F}^{eT}, \tag{3}$$

where $J^e = \det \boldsymbol{F}^e$. The strain energy function describes the material behavior of brain tissue mathematically. In our case, we consider a nonlinear hyperelastic material model as viscous effects, which have been observed for higher strain rates, become less relevant in the case of a slow process like brain development occurring over the course of weeks and months, as discussed in **Budday et al., 2020**. Our previous studies have shown that the isotropic neo-Hookean constitutive model best represents the material behavior of brain tissue during cortical folding (**Budday et al., 2020**). The corresponding strain energy function $\psi_g$ is given as

$$\psi_g(\boldsymbol{F}^e) = \frac{1}{2} \lambda \ln^2(J^e) + \frac{1}{2} \mu(r_i) \left[ \boldsymbol{F}^e : \boldsymbol{F}^e - 3 - 2\ln(J^e) \right], \tag{4}$$

where $\mu$ and $\lambda$ are the Lamé parameters. We use the nonlinear Heaviside function that is given in the general form as $\mathcal{H}(x; \gamma) = e^{\gamma x}/(1 + e^{\gamma x})$ to guarantee a smooth transition from the cortex to the subcortical plate with distinct mechanical parameters,

$$\mu(r_i) = \mu_s + \left[ \left[ \mu_c - \mu_s \right] \times \mathcal{H}(r_i - r_{\text{cp}}; 20) \right], \tag{5}$$

where the Heaviside function exponent $\gamma$ equals 20. Please note that we will later use a different value of $\gamma$ to serve numerical and geometrical requirements regarding the nature of the transition: higher values will lead to sharper transitions, lower values to smaller transitions. A more detailed discussion on the role of the value of $\gamma$ can be found in *Zarzor et al., 2021*. Our recent numerical simulation study suggested that the cortical stiffness continuously changes during human brain development due to the changes in the local microstructure (*Zarzor et al., 2021*). Accordingly, we formulate the cortical shear modulus $\mu_c$ as a function of the cell density,

$$\mu_c(c) = \begin{cases} \mu_\infty & \text{if} \quad c \geq c_{\text{max}}, \\ \mu_s + m_c(c - c_{\text{min}}) & \text{if} \quad c_{\text{max}} > c > c_{\text{min}}, \\ \mu_s & \text{if} \quad c \leq c_{\text{min}}. \end{cases} \tag{6}$$

It increases with increasing cell density in the range $\mu_c(c) \in [\mu_s, \mu_\infty]$, while the subcortical shear modulus $\mu_s$ remains constant. The slope is defined as $m_c = \mu_\infty - \mu_s/c_{\text{max}} - c_{\text{min}}$ and the stiffness ratio as $\beta_\mu = \mu_\infty/\mu_s$. Through *Equation 6*, the cell density problem controls the effective stiffness ratio between cortex and subcortex (as the cortical stiffness changes while the subcortical stiffness remains constant) and thus also the emerging cortical folding pattern (*Budday et al., 2014*; *Zarzor et al., 2021*).

## Mechanical growth problem

The growth tensor introduced in the Kinematics section is a key feature in our model that links the cell density problem with the mechanical problem. As it controls the amount and direction of growth, we need to consider how cellular processes affect the physiological growth behavior in order to find an appropriate formulation. During cellular migration, the subcortical layers expand isotropically. Then, under the effect of neuronal connectivity, the cortex grows – more pronounced in circumferential than in radial direction – as illustrated in *Figure 2*. Thus, we introduce the growth tensor as

$$\boldsymbol{F}^g = \vartheta^\perp \left[ \boldsymbol{I} - \boldsymbol{N} \otimes \boldsymbol{N} \right] + \vartheta^\| \boldsymbol{N} \otimes \boldsymbol{N}, \tag{7}$$

where $\boldsymbol{N}$ is the normal vector in the reference configuration $\mathcal{B}_0$ (it is linked to the spatial normal vector through $\boldsymbol{N} = \boldsymbol{F}^{-1} \cdot \boldsymbol{n}$), while $\vartheta^\perp$ and $\vartheta^\|$ denote the growth multipliers in circumferential and radial direction, respectively. Those multipliers control the amount of growth as a function of the cell density,

$$\vartheta^\perp = \left[ 1 + \kappa^\perp(r_i) \, c \right]^\alpha \qquad \text{and} \qquad \vartheta^\| = \left[ 1 + \kappa^\|(r_i) \, c \right]^\alpha, \tag{8}$$

where $\kappa^\perp$ and $\kappa^\|$ are the growth factors in the circumferential and radial direction, respectively, and $\alpha$ is the growth exponent. To ensure isotropic growth in the subcortical layers, we formulate those factors as a function of the radius $r_i$, such that

$$\kappa^\perp(r_i) = \kappa_s + \left[ \kappa_s \left[ \beta_\kappa - 1 \right] \times \mathcal{H}(r_i - r_{\text{cp}}; 20) \right] \quad \text{and} \tag{9}$$

$$\kappa^\|(r_i) = \kappa_s + \left[ \kappa_s \left[ \frac{1}{\beta_\kappa} - 1 \right] \times \mathcal{H}(r_i - r_{\text{cp}}; 20) \right], \tag{10}$$

where $\kappa_s$ is the growth factor in the subcortical layers, and $\beta_\kappa$ is the growth ratio between $\kappa^\perp$ and $\kappa_s$ (*Zarzor et al., 2021*). Through *Equation 8*, the amount of growth is directly related to the cell density – the higher the cell density, the more growth.

## Cell density problem

We formulate the balance equation of the cell density problem in such a way that we can mathematically describe the different cellular processes occurring at the microscopic scale. Temporal changes in

the cell density field are kept in balance by source and flux terms. The balance equation given in the spatial configuration $\mathcal{B}_t$ follows as

$$\frac{\dot{J}}{J} c + \dot{c} = -\boldsymbol{\nabla}_{\boldsymbol{x}} \cdot \left[ \hat{\mathbf{v}}(c, \boldsymbol{x}) \, c - \boldsymbol{d}^{cc}(\boldsymbol{x}) \cdot \boldsymbol{\nabla}_{\boldsymbol{x}} \, c \right] + r_1^c(\boldsymbol{x}, s) + r_2^c(\boldsymbol{x}, s), \tag{11}$$

where the first flux term $\hat{\mathbf{v}}(\boldsymbol{x}) \, c$ represents the migration in the subcortical plate, the second flux term $\boldsymbol{d}^{cc}(\boldsymbol{x}) \cdot \boldsymbol{\nabla}_{\boldsymbol{x}} \, c$ represents the neuronal connectivity in the cortex, the first source term $r_1^c(\boldsymbol{x}, s)$ represents cell proliferation in the VZ, and the second source term $r_2^c(\boldsymbol{x}, s)$ cell proliferation in the OSVZ. The migration velocity vector $\hat{\mathbf{v}}(\boldsymbol{x})$ guides the cells along radial glial cell fibers and controls their speed,

$$\hat{\mathbf{v}}(\boldsymbol{x}) = \mathcal{H}(c - c_0; \gamma_c) \, \mathrm{v}(r_i) \, \boldsymbol{n}/ \parallel \boldsymbol{n} \parallel . \tag{12}$$

The vector $\boldsymbol{n}$ represents the normalized orientation of radial glial cell fibers in the spatial configuration and controls the migration direction of neurons. As the brain grows and folds, the fiber direction changes. Through this feedback mechanism, the mechanical growth problem affects how neurons migrate and the cell density evolves locally. The nonlinear regularized Heaviside function $\mathcal{H}(c - c_0; \gamma_c)$ with the Heaviside exponent $\gamma_c$ links the migration speed with the cell density field. Accordingly, the cells start to migrate only when their density exceeds the critical threshold $c_0$. The value $\mathrm{v}$ specifies the maximum migration speed of each individual cell in the domain. To ensure that this value vanishes smoothly at the cortex boundary $r_{\mathrm{cp}}$, we formulate it as a function of the radial position $r_i$, as shown in **Figure 2B**, such that

$$\mathrm{v}(r_i) = \mathrm{v} \left[ 1 - \mathcal{H}(r_i - r_{\mathrm{cp}}; 10) \right] . \tag{13}$$

After the cells reach the cortex, they diffuse isotropically, as described by the diffusion tensor $\boldsymbol{d}^{cc}(\boldsymbol{x}) = (d^{cc}(r_i) + v_c(c)) \, \boldsymbol{I}$ with the diffusivity $d^{cc}$, the artificial viscosity $v_c(c)$, and the second order unit tensor $\boldsymbol{I}$. The artificial viscosity term $v_c(c)$ serves as a numerical stabilization to avoid numerical oscillations associated with the advection-diffusion equation. It only acts when the actual cell density does not satisfy the balance equation and ensures more reliable results without having a particular physical meaning. The diffusivity is defined as a function of the radial position $r_i$ to act only in the cortex,

$$d^{cc}(r_i) = d^{cc} \mathcal{H}(r_i - r_{\mathrm{cp}}; 10). \tag{14}$$

The first source term $r_1^c$ represents the radial glial cell proliferation in the VZ, as demonstrated in **Figure 2B**, and is given as

$$r_1^c(\boldsymbol{x}, s) = G_{\mathrm{vz}}^s(s) \left[ 1 - \mathcal{H}(r_i - r_{\mathrm{vz}}; 50) \right] \qquad \text{with} \tag{15}$$

$$G_{\{\bullet\}}^s(s) = G_{\{\bullet\}} - \begin{cases} (s - 1) \, G_{\{\bullet\}} & \text{if} \quad s < 1.8 \\ 0.8 \, G_{\{\bullet\}} & \text{else} \end{cases} \tag{16}$$

where $r_{\mathrm{vz}}$ is the outer radial boundary of the VZ and $G_{\mathrm{vz}}^s$ is the division rate in the VZ as a function of the maximum stretch $s$ in the domain. By applying **Equation 16** for the VZ, we ensure that the division rate decreases from its initial value $G_{\mathrm{vz}}$ to a smaller value as the maximum stretch value $s$ in the domain increases, i.e., with increasing gestational age. This constitutes an additional feedback mechanism between the mechanical growth problem and the cell density problem: As the maximum stretch and thus the deformation increases due to constrained cortical growth, the division rate in the VZ decreases, resulting in less newborn cells.

Besides the proliferation of radial glial cells around the cerebral ventricles in the VZ, the ORGCs proliferate in the OSVZ. To capture this effect, we add a second source term $r_2^c$, as demonstrated in **Figure 2B**. The second source term is given as

$$r_2^c(\boldsymbol{x}, s) = G_{\mathrm{osvz}}^s(s) \left[ \mathcal{H}(r_i - r_{\mathrm{isvz}}; 50) - \mathcal{H}(r_i - r_{\mathrm{osvz}}(t); 50) \right] , \tag{17}$$

where $r_{\mathrm{isvz}}$ is the outer radial boundary of the inner subventricular zone and $G_{\mathrm{osvz}}^s$ is the division rate in the OSVZ that again decreases with increasing maximum stretch $s$ in the domain. To numerically capture the expansion of the OSVZ under the effect of MST of ORGCs, we formulate the outer radial boundary of the OSVZ as a function of time, such that, $r_{\mathrm{osvz}} = r_{\mathrm{isvz}} + m_{\mathrm{mst}} \, t$, where $m_{\mathrm{mst}}$ is introduced

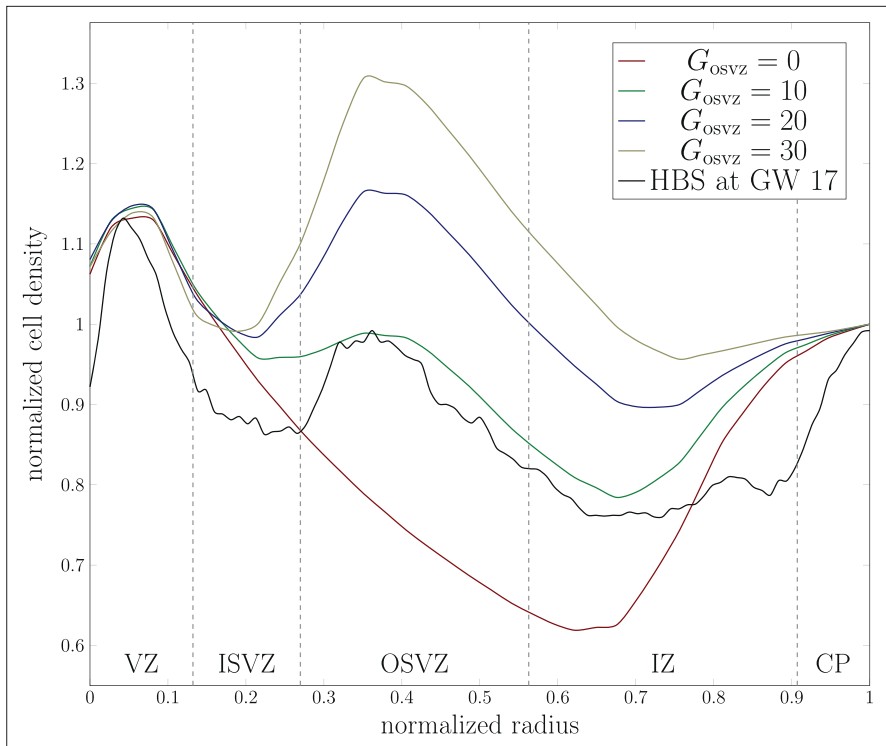

**Figure 4.** Evolution of the normalized cell density in the normalized radial direction from the ventricular surface to the outer cortical surface for numerical simulations and histologically stained human brain sections (HBS) at gestational week (GW) 17. The simulation results correspond to the varying cortical stiffness case with a stiffness ratio of 3, and a ventricular zone (VZ) division rate $G_{\text{VZ}} = 120$. The stained human brain sections results correspond to line L in *Figure 3*.

as the MST factor. Again, we apply *Equation 16* for the OSVZ, but in this case with the initial division rate $G_{\text{osvz}}$.

## Model parameters and boundary conditions

In this work, we will consider two different cases regarding the mechanical model: The first case considers a varying cortical stiffness as introduced in the Mechanical problem section, while the second case assumes a constant cortical stiffness, i.e., $\mu_c = \mu_\infty = \text{constant}$. While our previous study had suggested that the simulations with varying cortical stiffness lead to morphologies that better agree with those in the actual human brain (*Zarzor et al., 2021*), we still consider both cases in the following, varying stiffness and constant stiffness, as the situation might change when including the OSVZ and we aim to investigate corresponding interdependency effects. *Table 1* summarizes the model parameters that are used in the simulation for the 2D case. We will refer to the parameters changes in the 3D case later when we present the corresponding results to avoid confusion. The mechanical and diffusion parameters are adapted from the literature (*Budday et al., 2020*; *de Rooij and Kuhl, 2018*), while the geometry parameters are estimated based on histologically stained human brain sections and magnetic resonance images. For instance, to determine the MST factor, we measured the relative distance between the inner subventricular zone and OSVZ in histologically stained images. The final value adopted is the result of dividing the measured distance by the expected time. When determining the growth problem parameters, numerical stability and algorithm convergence were major criteria.

We have previously thoroughly studied the effect of the stiffness ratio on the resulting folding pattern (*Zarzor et al., 2021*). Here, we choose a stiffness ratio of 8 for the constant stiffness case and a ratio of 3 for the varying stiffness case. Those values led to the best agreement of simulation results with data from stained histological sections regarding the local gyrification index value and the thickness ratio between gyri and sulci. For more details, we refer to *Zarzor et al., 2021*. We note that

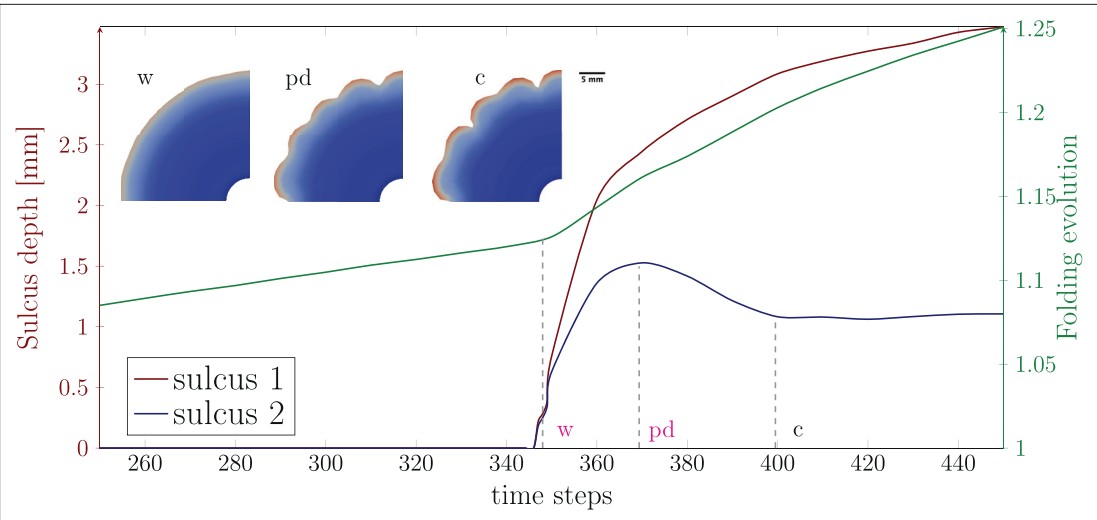

**Figure 5.** Primary and secondary mechanical instabilities in the developing brain. The temporal courses of the depth of two sulci (red and blue curve) and the folding evolution (green curve), as denoted in *Figure 6*, indicate the mechanical instability points. Initially, the brain surface is smooth, the sulcus depth is zero, and the folding evolution increases only gradually. At the first instability point, the cortex starts to fold into wrinkles (**w**), where the sulci deepen uniformly. Due to the transition from a smooth to a wrinkled surface, the folding evolution now shows a more rapid increase. At the second instability point, a pitchfork-like bifurcation occurs, where every second sulcus continues to deepen while those in between become shallower. This results in a period doubling (pd) pattern, which is well visible and fully established at state c. The results correspond to the varying stiffness case with $G_{vz} = 120$, $G_{osvz} = 20$, and a stiffness ratio $\beta_\mu = 3$.

the tissue shows a stiffer behavior in the case of constant stiffness than in the case of varying stiffness for the same value of the stiffness ratio. For that reason, a higher stiffness ratio (lower stiffness in the subcortical layers since the final cortical stiffness $\mu_\infty$ is constant in both cases) is required in the case of constant stiffness to achieve a similar level of folding.

Finally, we use homogenized Dirichlet boundary conditions on the inner brain surface and homogenized Neumann boundary conditions on the outer surface. Furthermore, in the 2D case, we constrain the right edge in the x direction and the bottom edge in the y direction. In the 3D case, we constrain

**Table 1.** Model parameters in the two-dimensional case.

| Geometry parameters | | | | Cell density problem parameters | | | |
|---|---|---|---|---|---|---|---|
| **Parameter** | | **Value** | **Unit** | **Parameter** | | **Value** | **Unit** |
| Outer brain radius | $R$ | 2 | mm | Division rate in VZ | $G_{vz}$ | [30-120] | mm$^{-2}$d$^{-1}$ |
| Inner brain radius | $r$ | 0.4 | mm | Division rate in OSVZ | $G_{osvz}$ | [10-30] | mm$^{-2}$d$^{-1}$ |
| VZ radius | $r_{vz}$ | 0.5 | mm | Migration speed | v | 5 | mm d$^{-1}$ |
| ISVZ radius | $r_{isvz}$ | 0.8 | mm | Migration threshold | $c_0$ | 500 | mm$^{-2}$ |
| Cortex radius | $r_{cp}$ | 1.8 | mm | Heaviside exponent | $\gamma_c$ | 0.008 | – |
| MST factor | $m_{mst}$ | 0.02 | mm d$^{-1}$ | Diffusivity | $d^{cc}$ | 0.11 | mm$^{-2}$d$^{-1}$ |
| Mechanical problem parameters | | | | Mechanical growth problem parameters | | | |
| **Parameter** | | **Value** | **Unit** | **Parameter** | | **Value** | **Unit** |
| Cortex shear modulus | $\mu_\infty$ | 2.07 | kPa | Growth parameter | $\kappa_s$ | $4.07e^{-4}$ | mm$^2$ |
| Poisson ratio | $\nu$ | 0.38 | – | Growth exponent | $\alpha$ | 1.65 | – |
| Stiffness ratio | $\beta_\mu$ | 3,8 | – | Growth ratio | $\beta_\kappa$ | 1.5,3 | – |
| Maximum threshold | $c_{max}$ | 700 | mm$^{-2}$ | | | | |
| Minimum threshold | $c_{min}$ | 200 | mm$^{-2}$ | | | | |

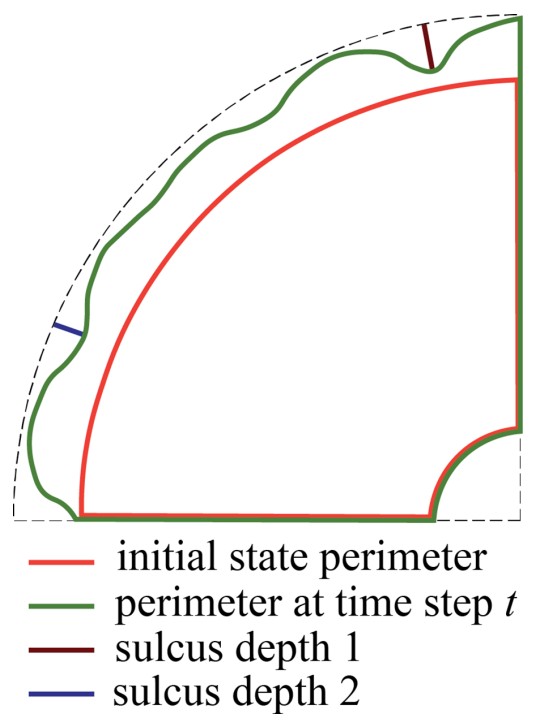

**Figure 6.** Quantification of the depth of two sulci (sulcus 1 and sulcus 2) and the folding evolution, defined as the ratio between the outer perimeter at time step *t* and the initial perimeter.

the bottom surface in the y direction and keep it free to extend in the x and z directions. For more details about boundary conditions and numerical details, we refer to *Zarzor et al., 2021*.

## Model validation

We validate our computational model by comparing the simulation results with histologically stained sections of the human fetal brain. For details on the corresponding preparation and staining, we refer to *Zarzor et al., 2021*. The sections belong to human fetuses aborted at gestational weeks 17, 24, 30, and 34. After defining the areas representative of the simulation domain introduced in the previous section (see *Figure 2A*) for each gestational week, we detect and assess the cell density using the software *Qupath*. This study was approved by the ethics review board of the Friedrich-Alexander-Universität Erlangen-Nürnberg with the reference number 22-209-Bp, and all procedures were conducted in accordance with the Declaration of Helsinki. In the following, we summarize the steps we followed to determine the cell density in human fetal brain sections. Since this analysis is intended to mainly serve as a means of comparison between different gestational weeks, a more detailed analysis using individual markers was not necessary at this stage.

1. Identify regions of interest to which the analysis should be applied, as shown in *Figure 3A*.
2. Pre-process the annotated area to ensure better cell detection through image color transfer, contrast ratio modification, and stain vector settings.
3. Automatically detect the cells in the relevant area by using the 'positive cell detection' command in *Qupath*, which distinguishes between cell types according to the staining. Here, the negative cells (blue) are mostly glial cells or extracellular matrix components, while positive cells (red) are mostly neurons or progenitor cells, as shown in *Figure 3B*. While this distinction between cell types might not be fully accurate, the results are satisfactory for our specific application.
4. Count the nearby detections in a small circle with an arbitrarily chosen radius $\epsilon = 100\,\mathrm{m}$ around each detected positive cell (i) to determine the corresponding cell density by dividing the number of nearby detections by the circular area, as demonstrated in *Figure 3C*.
5. Visualize the cell density, as illustrated in *Figure 3A*.

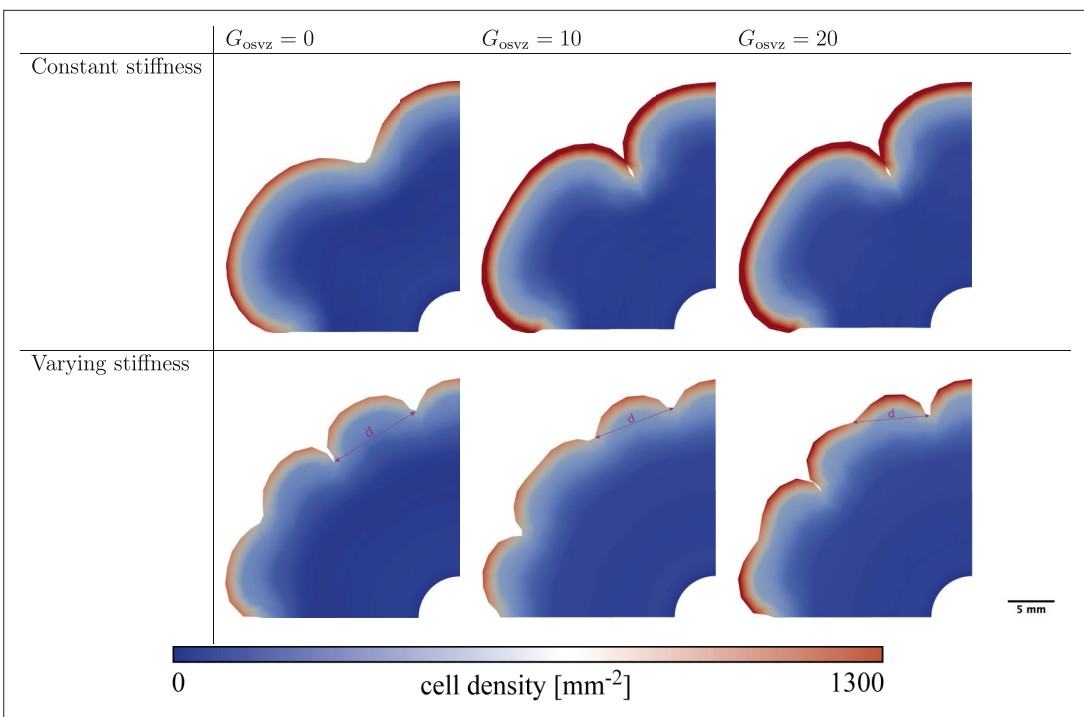

**Figure 7.** Final folding patterns at gestational week 36 for different values of the division rate in the outer subventricular zone (OSVZ) $G_{osvz}$ for the constant (top) and varying (bottom) cortical stiffness cases. The remaining parameters are fixed as follows: division rate in the ventricular zone (VZ) $G_{vz} = 120$, stiffness ratio $\beta_\mu = 8$ for constant stiffness, and $\beta_\mu = 3$ for varying stiffness. The marked distance between sulci $d$ decreases with increasing $G_{osvz}$.

*Figure 3* shows the cell density distribution around gestational week 17 and demonstrates the densely packed VZ due to the high proliferation rate of radial glial cells, with about $13,600\,\text{cells/mm}^2$. In addition, we can locate the other zones introduced in *Figure 1*. The OSVZ shows a higher cell density (approximately $9550\,\text{cell/mm}^2$) than both the inner subventricular and the intermediate zone. This zone is composed of several types of cells, the original intermediate progenitor cells that migrated from the VZ, migrated neurons, ORGCs that produce more intermediate progenitor cells, and newborn neurons that are produced through the intermediate progenitor cells' asymmetric division. Our analysis of the histologically stained sections of the human fetal brain (see *Figure 3*) also shows that the OSVZ is 1.5 times thicker than the inner subventricular zone, even though it did not emerge at an earlier stage of development (the inner subventricular zone emerges around gestational week 7 and the OSVZ around gestational week 11). This implies that the thickness of the OSVZ increases with time. The intermediate zone is characterized by a low cell density with about $1600\,\text{cells/mm}^2$. Still, it is a transit area for the migrating neurons. The higher cell density in the inner subventricular zone $4800\,\text{cells/mm}^2$ corroborates the presence of another type of cell besides migrating neurons, i.e., intermediate progenitor cells. The migration process in gestational week 17 is still ongoing – the cortex is not yet fully developed and filled with neurons with only about $12,000\,\text{cells/mm}^2$. Therefore, at this stage of development, the VZ still has the highest cell density in the brain. *Figure 4* compares the cell density distribution along a line from the ventricular surface to the outer cortical surface in the human fetal brain at gestational week 17 (indicated by line L in *Figure 3*) with different simulation results. For better comparability and to avoid differences in the dimensions between the histologically stained sections of the human fetal brain and the simulation domain, we normalize the domain's radius according to the extension from the ventricular to the outer cortical surface in the stained sections. In addition, we normalize the cell density with respect to its maximum value in the cortex. According to our previous work (*Zarzor et al., 2021*), we include a varying stiffness in the cortex during human brain development in our simulations, adopt a stiffness ratio of 3, and a division rate in the VZ of 120.

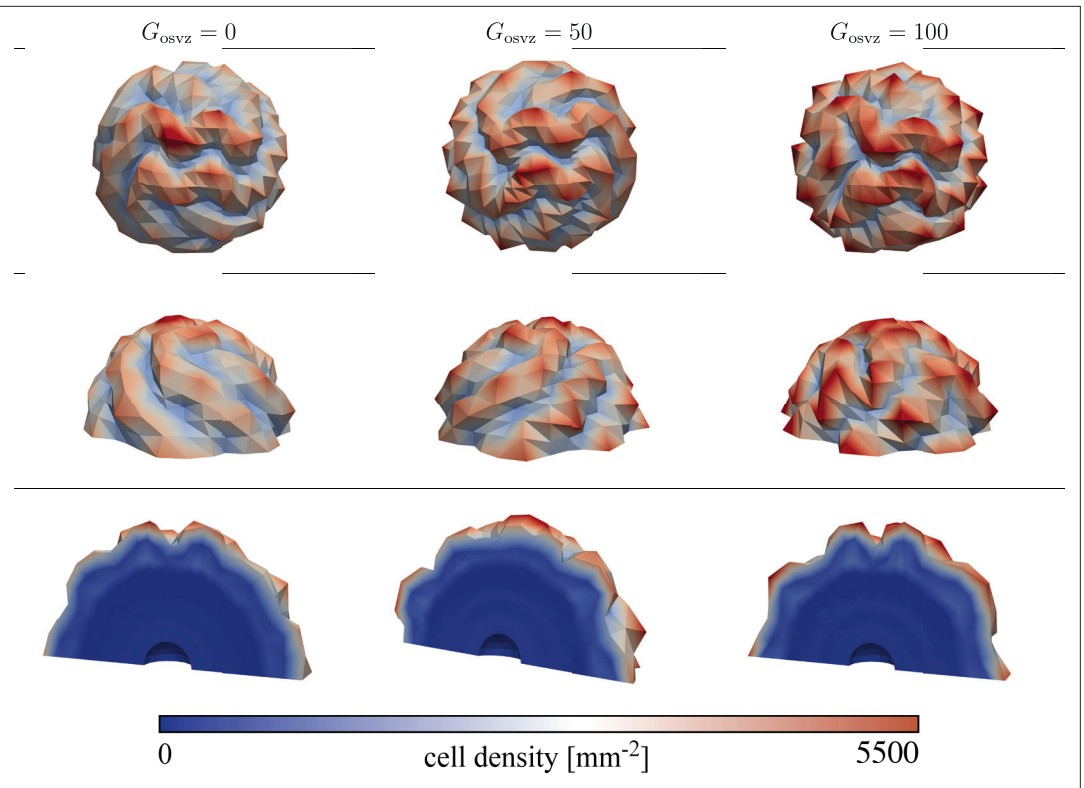

**Figure 8.** Final folding patterns at gestational week 36 for the hree-dimensional (3D) model with varying cortical stiffness, a stiffness ratio of 3, a growth ratio of 3, and an initial division rate in the ventricular zone (VZ) $G_{vz} = 600$. The folding complexity increases with increasing initial division rate in the outer subventricular zone (OSVZ) $G_{osvz}$.

The simulation results for an initial division rate in the OSVZ of $G_{osvz} = 10$ well capture the trends observed in the histologically stained sections of the human fetal brain. The cell density shows a first local peak representing the VZ for a normalized radius of approximately 0.05. It then gradually decreases to reach its first local minimum in the inner subventricular zone for a normalized radius of 0.3. This effect is less pronounced in the simulations than in the actual human fetal brain. The curves start to rise again in the OSVZ to reach the second peak for a normalized radius of 0.4. Again, the simulation results capture this peak quite accurately, with a difference of only 3%. The second local minimum represents the intermediate zone for a normalized radius of 0.7, while the third peak represents the cortex. The simulation results for $G_{osvz} = 20$ and 30 result in a higher cell density in the OSVZ than in the actual human fetal brain. In contrast, the curve for $G_{osvz} = 0$ (i.e., without including the effect of the OSVZ) shows a significantly decreased cell density in the OSVZ and intermediate zone.

## Results and discussion

In this section, we apply our computational model to answer some of the major questions regarding the role of the OSVZ for cortical folding during human brain development. As mentioned above, we consider both cases, constant and varying cortical stiffness. Before addressing the individual questions, we would like to first highlight some general features of the mechanical instability problem as the underlying mechanism of cortical folding. *Figure 5* shows an exemplary temporal course of both sulcus depth and folding evolution – defined as the ratio between the outer perimeter at time step $t$ and the initial perimeter (see *Figure 6*) – for the varying stiffness case with $G_{vz} = 120$ and $G_{osvz} = 20$. The outer brain maintains its smooth surface until the first instability point, where compressive stresses in the cortex induced by cell-density-driven cortical growth reach a critical value, the cortex starts to fold and sulci deepen rapidly and uniformly. At the instability point, the curve of the folding evolution shows a kink, as the outer brain surface now expands more rapidly. After a second instability point,

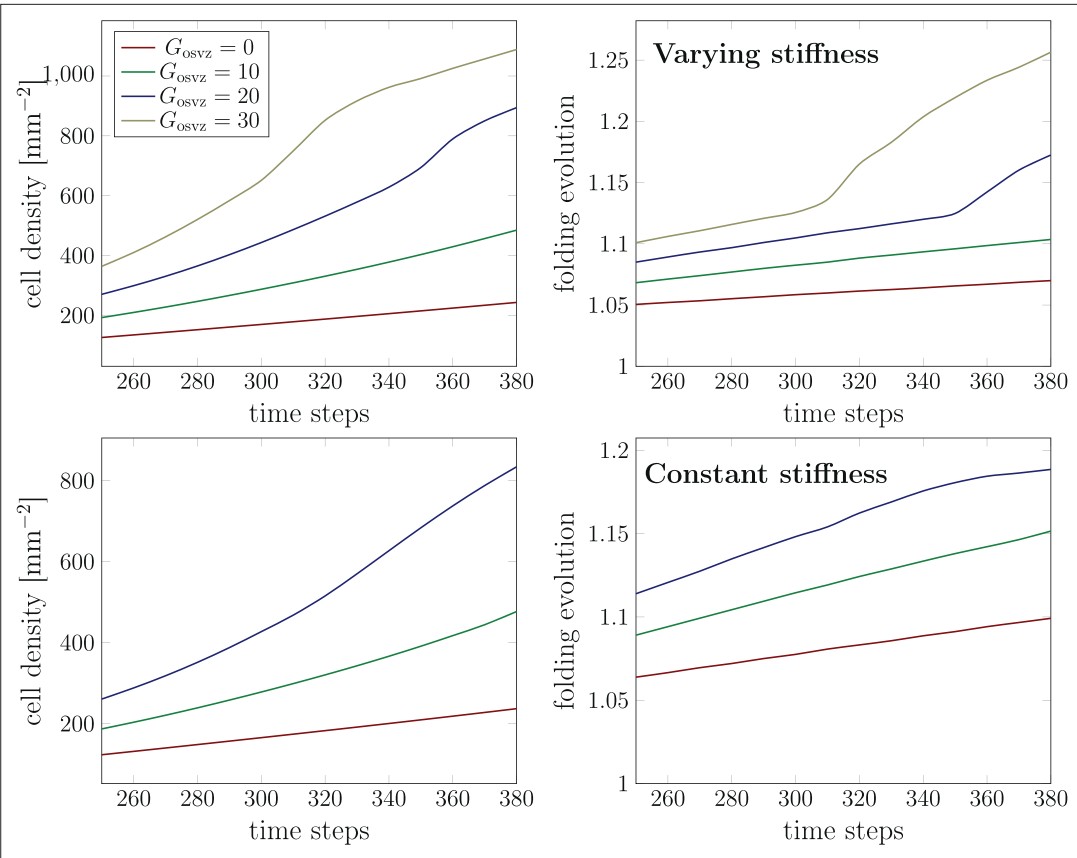

**Figure 9.** Temporal evolution of the maximum cell density and the folding evolution (the current outer perimeter divided by the initial perimeter, as indicated in *Figure 6*) at a constant division rate in the ventricular zone (VZ) $G_{VZ} = 120$ and different initial division rates in the outer subventricular zone (OSVZ) $G_{OSVZ}$. The results in the top row correspond to the varying cortical stiffness case with a stiffness ratio of 3. The results in the bottom row correspond to the constant cortical stiffness case with a stiffness ratio of 8.

where a secondary instability occurs and we see a pitchfork-like bifurcation in *Figure 5*, only every second sulcus continues to deepen, while those in between become shallower again. We refer to the resulting folding pattern as period doubling pattern. For a detailed discussion of secondary instabilities, period doubling, and even period tripling patterns and their role in cortical folding, we refer to our previous works in *Budday et al., 2015a* and *Budday et al., 2015c*.

## How does cell proliferation in the OSVZ affect cortical folding patterns?

Many previous studies have tried to address this point (*Hansen et al., 2010*). However, through purely experimental approaches, it is difficult to answer this question. In our model, we can apply different values of the division rate in the OSVZ ($G_{OSVZ}$) to show how ORGC proliferation affects cortical folding. *Figure 7* shows the folding patterns emerging at gestational week 36 for both varying and constant stiffness cases and different values of the initial division rate in the OSVZ $G_{OSVZ}$. In the case of constant stiffness, the effect is marginal. In the case of varying stiffness, there is a more noticeable change in the folding patterns. In general, the distance between neighboring sulci decreases with increasing $G_{OSVZ}$, as marked in *Figure 7*. For the displayed cases, the distance decreases from $d = 8.796$ mm for $G_{OSVZ} = 0$ to $d = 8.67$ mm for $G_{OSVZ} = 10$ and $d = 8.2$ mm for $G_{OSVZ} = 20$. Interestingly, the cortical thickness and effective stiffness ratio at the first instability point (denoted by w in *Figure 5*) are the same for all these cases. Therefore, we attribute the observed differences to the faster increase in the cell density and thus cortical growth, cortical stiffness, and the effective stiffness after the instability has been initiated. In addition, we observe period doubling patterns emerge (*Budday et al., 2015a*; *Budday et al.,*

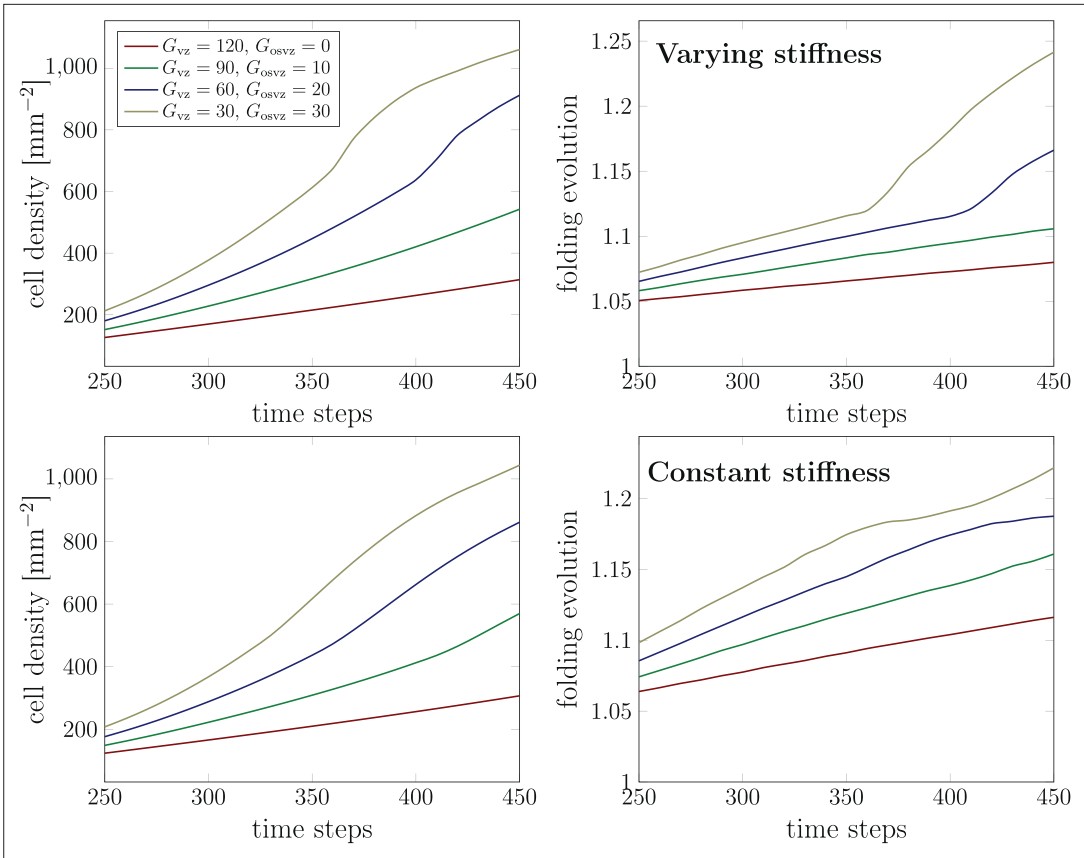

**Figure 10.** Temporal evolution of the maximum cell density and the folding evolution for different division rates in the ventricular zone (VZ) $G_{vz}$ and outer subventricular zone (OSVZ) $G_{osvz}$. The results in the top row correspond to the varying cortical stiffness case with a stiffness ratio of 3. The results in the bottom row correspond to the constant cortical stiffness case with a stiffness ratio of 8.

---

2015c), which are most pronounced at a value of $G_{osvz} = 20$. This indicates that the proliferation in the OSVZ enhances secondary mechanical instabilities and leads to more complex folding patterns earlier.

*Figure 8* demonstrates that the observed trends also hold true when extending the model to 3D. For the case of varying stiffness with a stiffness ratio of 3, a growth ratio of 3, and an initial division rate in the VZ $G_{vz} = 600$, the folding complexity increases with increasing initial division rate in the OSVZ $G_{osvz}$.

Besides the direct relation between the proliferation in the OSVZ and the folding morphology, there are indirect effects and other aspects concerning ORGC proliferation, which will be discussed in more detail in the following sections.

## How does cell proliferation in the OSVZ affect the cell density and folding evolution?

After we have assessed the effect of cell proliferation in the OSVZ on the final folding pattern, we investigate its effect on the evolution of both cell density and folding morphology. *Figure 9* shows the temporal evolution of the maximum cell density in the domain and the folding evolution between time steps 260 and 380 for different initial division rates in the OSVZ $G_{osvz}$ and a constant division rate in the VZ $G_{vz} = 120$. Again, we consider both cases, constant and varying cortical stiffness. Increasing the initial division rate in the OSVZ leads to a significant increase in both the cell density and folding evolution. Consequently, for the case of $G_{osvz} = 30$, the cell density reaches the highest value of $1100\,\text{mm}^{-2}$, corresponding to a folding evolution of 1.25. For the case of $G_{osvz} = 0$, in contrast, the cell density does not even exceed the migration threshold. Comparing the cases $G_{osvz} = 20$ and $G_{osvz} = 30$ shows that the instability occurs earlier with increasing initial division rate in the OSVZ $G_{osvz}$. Thus, the ORGC proliferation decreases the time required to reach the final folding pattern. In general, the differences

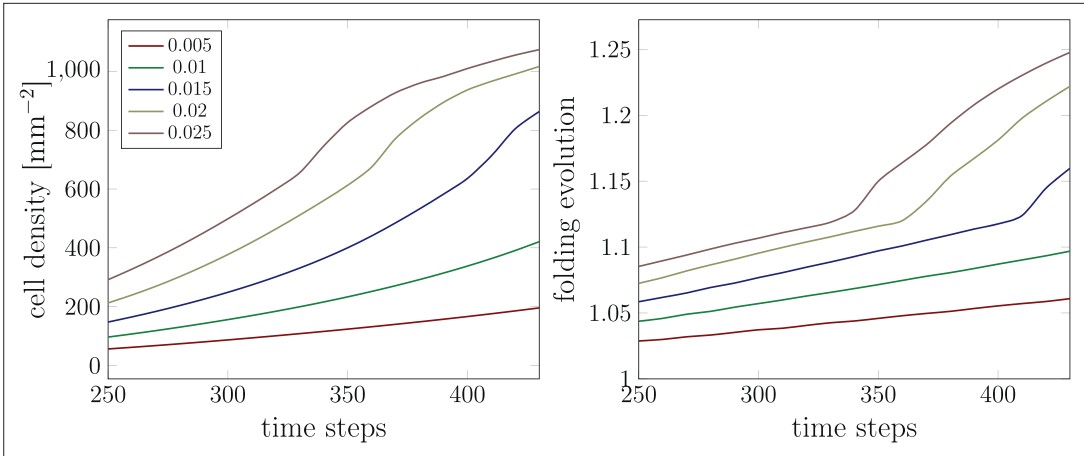

**Figure 11.** Temporal evolution of the maximum cell density and the folding evolution for different values of the mitotic small translocation (MST) factor. The results correspond to the varying cortical stiffness case with a stiffness ratio of 3, a division rate in the ventricular zone (VZ) $G_{vz} = 30$, and an initial division rate in the outer subventricular zone (OSVZ) $G_{osvz} = 30$.

between the results for the constant and varying cortical stiffness case are minor. However, the curves for the varying cortical stiffness case rise faster than for the constant case. We note that we could not generate results for the initial division rate in the OSVZ $G_{osvz} = 30$ and the constant cortical stiffness case due to numerical issues.

## Which proliferation zone is more influential for fetal human brain development?

To answer the question which one of the proliferation zones (VZ or OSVZ) is more important for cellular brain development and cortical folding, we have implemented four sets of division rates. The first set assumes a high initial division rate in the VZ $G_{vz}$ and a low initial division rate in the OSVZ $G_{osvz}$. For the following parameter sets, we gradually decrease $G_{vz}$, while we increase $G_{osvz}$. *Figure 10* shows the corresponding results for the maximum cell density in the domain and the folding evolution between time steps 250 and 450 for both cases, constant and varying cortical stiffness. Unexpectedly, the cell density for the set ($G_{vz} = 30$, $G_{osvz} = 30$) rises faster than the one for the set ($G_{vz} = 120$, $G_{osvz} = 0$). Indeed, not only the cell density is affected by increasing the division rate in the OSVZ (at the expense of decreasing it in the VZ), but also convolutions (cortical folds) appear earlier, as illustrated by the curve for the folding evolution (*Figure 10*, right). Our results thus indicate an unproportionally strong effect of ORGC proliferation in the OSVZ on cellular brain development and cortical folding. We attribute this observation to the larger volume occupied by the OSVZ compared to the VZ. Concerning

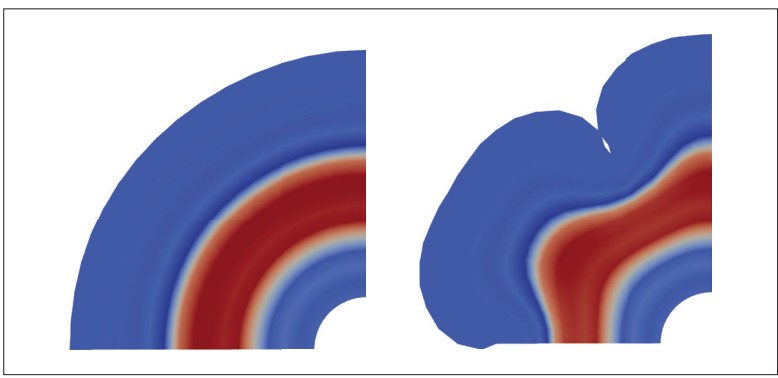

**Figure 12.** The effect of cortical folding on the outer subventricular zone (OSVZ). While the OSVZ has a constant thickness before cortical folds emerge (left), it later becomes thicker beneath gyri than beneath sulci (right).

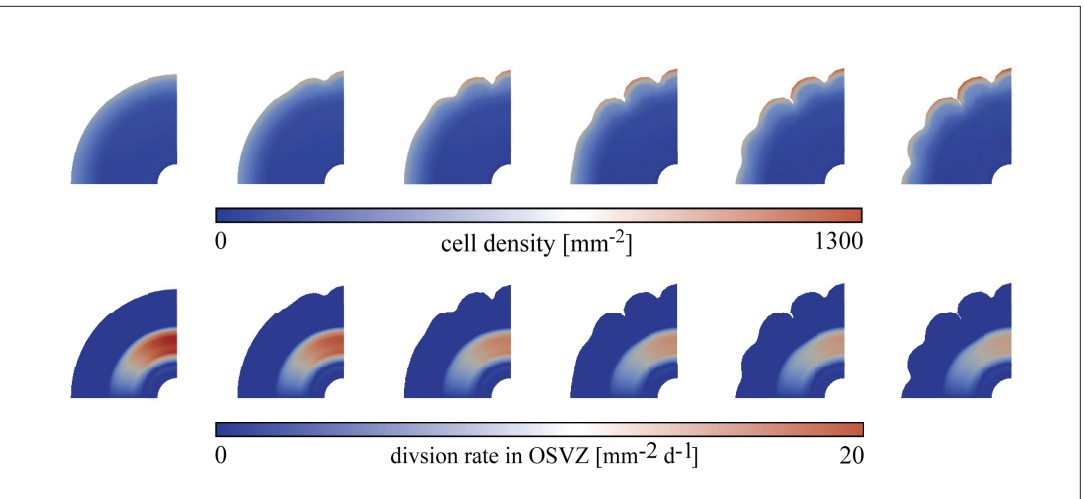

**Figure 13.** Temporal evolution of cortical folds for a gradually decreasing outer subventricular zone (OSVZ) division rate $G_{osvz}$ along the circumferential direction between time steps 325 and 450 for the varying cortical stiffness case, an initial division rate in the ventricular zone (VZ) $G_{VZ} = 120$, and varying division rates in the OSVZ with an initial value of 20. The top row shows the cell density distribution and the bottom row the actual division rate in the OSVZ.

the difference between the results for the constant and varying cortical stiffness case, the curves for a varying stiffness rise faster than for a constant stiffness.

## How does the MST behavior of ORGCs affect brain development?

We have introduced a specific simulation parameter to capture the MST behavior of ORGCs, which now allows us to study the effect of the MST factor on cortical folding. For this parameter study, we limit ourselves to the varying cortical stiffness case with an initial division rate in the VZ $G_{VZ} = 30$ and an initial division rate in the OSVZ $G_{osvz} = 30$. *Figure 11* shows the temporal evolution of the maximum cell density and the folding evolution between time steps 250 and 430 for different values of the MST factor. Our simulations indicate that with increasing MST factor, the value of the maximum cell density in the domain increases exponentially. Since the MST factor incorporates the expansion of the

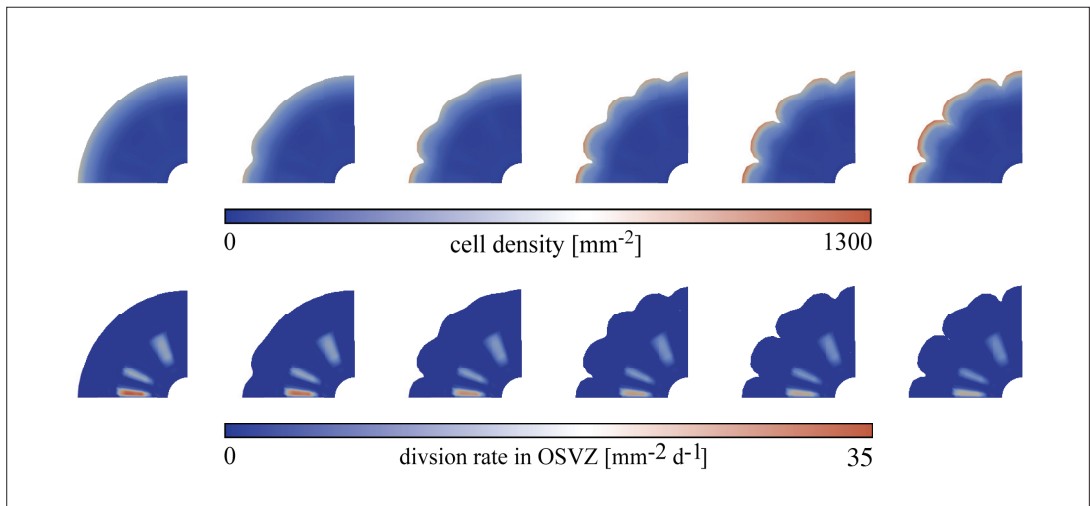

**Figure 14.** Temporal evolution of cortical folds for a random distribution of the outer subventricular zone (OSVZ) division rate $G_{osvz}$ between time steps 450 and 670 for the varying cortical stiffness case, an initial division rate in the ventricular zone (VZ) $G_{VZ} = 120$, and varying division rates in the OSVZ with an initial value of 20. The top row shows the cell density distribution and the bottom row the actual division rate in the OSVZ.

OSVZ with time, these results are consistent with the observation in the previous section, showing an increasing spatial expansion of the OSVZ.

## Is the OSVZ affected by cortical folding?

After we have discussed the effect of the OSVZ on the formation of cortical folds, we will now discuss the opposite – the effect of cortical folding on the OSVZ. Although the OSVZ has a constant thickness in our model throughout the entire domain in the intermediate stage of human brain development (before cortical folds emerge), as demonstrated in red in *Figure 12*, left, the simulations show that this quickly changes after the first folds start to emerge. The thickness of the OSVZ starts to show spatial variations and becomes thicker beneath gyri and thinner beneath sulci, as shown in *Figure 12*, right. This result is consistent with what was previously observed experimentally in the human brain (***Kostović et al., 2002***). While it is to date not clear whether this phenomenon is rather the cause or the result of cortical folding, our study clearly indicates that the (mechanics-driven) process of cortical folding is sufficient to induce OSVZ variations. The forces generated by the mechanical growth problem not only fold the cortical layer but also lead to undulations in the deeper zones. Still, the deeper proliferating zones (i.e., inner subventricular zone and VZ) remain equally smooth as the ventricular surface.

## Are cortical folds affected by regional proliferation variations in the OSVZ?

Previous studies have emphasized the existence of variations in the ORGC proliferation rate in different brain regions (***Hansen et al., 2010***). Some of those have suggested that the proliferation rate is higher beneath gyri than beneath sulci (***Borrell, 2018***). Similarly, in the ferret brain, where a region close in structure to the primate's OSVZ was found, this region shows a unique mosaic-like structure (***Fietz et al., 2010***; ***Reillo and Borrell, 2012***). In this section, we aim to assess the effect of regional proliferation variations in the OSVZ on the emerging cortical folding pattern. We discuss two different heterogeneous patterns here, but have included more variations online through our user interface on GitHub, as described in the Data availability section. In the first case, the OSVZ division rate gradually decreases along the circumferential direction. In the second case, the division rate varies in a more random pattern. *Figures 13 and 14* show how cortical folds develop in both cases for the varying cortical stiffness case, an initial division rate in the VZ $G_{vz} = 120$, and an initial division rate in the OSVZ $G_{osvz} = 20$. As expected, the evolving folding patterns slightly differ. In both cases, the first folds appear where the cell proliferation rate is highest. As expected, those regions also show a higher cell density in the cortex than regions nearby. However, both cases lead to final patterns with similar distances between sulci and folding complexity (one period doubling pattern). In addition, gyri and sulci are distributed equally – regardless of the division rate. Therefore, we may conclude that inhomogeneous cell proliferation in the OSVZ controls the location of first gyri and sulci but does not necessarily affect the distance between sulci (also referred to as folding wavelength) and the overall complexity of the emerging folding pattern. This agrees well with our previous finding that the characteristic wavelength of folding remains relatively stable for inhomogeneous cortical growth patterns (***Budday and Steinmann, 2018***). The simulation results are also consistent with the previously found remarkable surface expansion above the regions with higher proliferation in the OSVZ (***Llinares-Benadero and Borrell, 2019***).

## Conclusion

In this work, we have made use of a computational model for brain growth to provide insights into the role of the OSVZ – a unique additional proliferating zone in humans – during cortical folding in the developing brain. By using computational tools, we have addressed different open questions with the aim of valuably supplementing classical experimental approaches. The advantage of the computational model is that we can, on the one hand, clearly isolate purely physical mechanisms. On the other hand, it is possible to systematically study and understand the effect of individual parameters, which is much more difficult through in vitro and in vivo experiments. Our simulations have demonstrated how the proliferation in the OSVZ enhances the complexity of cortical folding patterns in 2D and 3D. Our results show that the existence of the OSVZ particularly triggers the emergence of secondary mechanical instabilities leading to more complex folding patterns. Furthermore, the proliferation

of ORGCs reduces the time required to induce the mechanical instability and thus cortical folding. Interestingly, our simulation results suggest that the generated mechanical forces not only 'fold' the cortex but also deeper subcortical zones including the OSVZ, which becomes thicker beneath gyri and thinner beneath sulci as a result of cortical folding. Consequently, our analyses suggest that the purely mechanics-driven process of cortical folding is sufficient to induce regional differences in the thickness of the OSVZ. Finally, our simulations reveal that inhomogeneous cell proliferation patterns in the OSVZ can control the location of first gyri and sulci but do not necessarily affect the distance between sulci and the overall complexity of the emerging folding pattern.

In conclusion, our physics-based computational modeling approach has allowed us to systematically assess the role of the OSVZ during human brain development and its effect on cortical folding – the classical hallmark of the human cortex at the organ scale. In the future, the computational framework can be used to not only better understand physiological brain development but also pathological processes – especially those involving abnormal cortical folding patterns. The computational model is able to shed new light on the interplay between the multiple processes at different scales and can help identify the main controlling parameters. However, it can only complement, not substitute, sophisticated experimental approaches that are still needed to answer questions, e.g., regarding the functional difference between progenitor cell types and corresponding lineage decisions.

## Acknowledgements

We would like to cordially thank Bettina Seydel for digitalizing the histological sections. In addition, we gratefully acknowledge the funding by the Deutsche Forschungsgemeinschaft (DFG, German Research Foundation) through the grant BU 3728/1-1 to SB and project number 460333672 – CRC 1540 Exploring Brain Mechanics (subprojects A01 and A02) to SB and IB.

## Additional information

### Funding

| Funder | Grant reference number | Author |
| --- | --- | --- |
| Deutsche Forschungsgemeinschaft | BU 3728/1-1 | Silvia Budday |
| Deutsche Forschungsgemeinschaft | project number 460333672 - CRC 1540 Exploring Brain Mechanics (subproject A01) | Silvia Budday |
| Deutsche Forschungsgemeinschaft | project number 460333672 - CRC 1540 Exploring Brain Mechanics (subproject A02) | Ingmar Blumcke |

The funders had no role in study design, data collection and interpretation, or the decision to submit the work for publication.

### Author contributions

Mohammad Saeed Zarzor, Conceptualization, Software, Validation, Investigation, Visualization, Methodology, Writing - original draft; Ingmar Blumcke, Resources, Validation, Writing - review and editing; Silvia Budday, Conceptualization, Supervision, Funding acquisition, Methodology, Writing - original draft, Project administration, Writing - review and editing

### Author ORCIDs

Mohammad Saeed Zarzor ⓘ http://orcid.org/0000-0002-3005-6115
Silvia Budday ⓘ http://orcid.org/0000-0002-7072-8174

### Ethics

This study was approved by the ethics review board of the Friedrich-Alexander-Universität Erlangen-Nürnberg with the reference number 22-209-Bp. Informed consent and consent to publish was obtained and all procedures were conducted in accordance with the Declaration of Helsinki.

### Decision letter and Author response

Decision letter https://doi.org/10.7554/eLife.82925.sa1
Author response https://doi.org/10.7554/eLife.82925.sa2

## Additional files

### Supplementary files

• Transparent reporting form

### Data availability

The datasets and code generated and/or analyzed during the current study are available on GitHub with a user interface to change model parameters and run simulations: https://github.com/SaeedZarzor/BFSimulator (copy archived at swh:1:rev:5fd9298c69a446b5ff21fff2ecaff5ef8a0ac085).

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
