## [Editor Report]

Through theoretical analysis, the authors argue that the proliferation of neurons in the outer subventricular zone, which is specific to humans, decreases the distance between neighboring sulci in the cerebral cortex and increases cell density in the ventricular zone. Though the exact mechanisms remain to be further elucidated, the compelling data and approach represent a valuable foundation for the study of cortical folding from the underpinning cellular level as well as the coupling role of mechanics and cellular biology. This study will be of particular interest to the large community of scientists studying the mechanisms of brain development and disorder and even possibly beyond.

---

## [Decision Letter]

**Decision letter after peer review:**

Thank you for submitting your article "Exploring the role of the outer subventricular zone during cortical folding through a physics-based model" for consideration by *eLife*. Your article has been reviewed by 3 peer reviewers, one of whom is a member of our Board of Reviewing Editors, and the evaluation has been overseen by Aleksandra Walczak as the Senior Editor. The following individuals involved in the review of your submission have agreed to reveal their identity: Jean-Francois Mangin (Reviewer #2); Xianqiao Wang (Reviewer #3).

Essential revisions:

Several assumptions underlying the model need to be discussed in more detail.

1) Clarify what is new in the model from Zarzor et al., 2021.

2) Comment on why there is no feedback from mechanics on cell division, migration, and diffusion.

3) Section on "How does cell proliferation in the OSVZ affect cortical folding patterns?" Clearly point out the differences between the panels in Figure 5. Add a quantification of the distance between sulci (maybe for different parameter values, you get a clearer readout). Also, comment on the mechanism underlying the decrease in distance.

4) Please, elaborate a little on the possibility of extending the model in 3D.

5) Ll. 456: "While it is to this date not clear whether this phenomenon (thickness variation of the OSVZ) is rather the cause or the result of cortical folding, our study clearly supports the latter." Since, in your model, there is no mechanism that would lead to thickness variation of the OSVZ, you can probably not make this statement. Please, clarify. However, you could state that cortical folding is sufficient to induce OSVZ thickness variations, right?

*Reviewer #1 (Recommendations for the authors):*

Why is there no feedback from mechanics on cell division?

We apologize for the lack of clarity. The division rate in both the VZ and the OSVZ is indeed controlled by the mechanical deformation and decreases with increasing maximum stretch s, Equation 16.

P. 11: "How does cell proliferation in the OSVZ affect cortical folding patterns?" I find this section rather disappointing. Figure 5 shows hardly any difference between the folds in the presence or absence of proliferation in the OSVZ. The distance between sulci is not quantified. Maybe for different parameter values, you get a clearer readout. Also, the mechanism underlying the decrease in distance is not explained. This makes this analysis look very superficial.

How does the presence of the OSVZ affect 3d folding patterns? Clearly, this goes beyond the work presented in this study but is the question we are really interested in. At least you should argue, why the 2d analysis is informative about the 3d case.

Ll. 456: "While it is to this date not clear whether this phenomenon (thickness variation of the OSVZ) is rather the cause or the result of cortical folding, our study clearly supports the latter." Since, in your model, there is no mechanism that would lead to thickness variation of the OSVZ, you cannot make this statement, I would say. You can state that cortical folding is sufficient to induce OSVZ thickness variations.

*Reviewer #2 (Recommendations for the authors):*Suggestions to authors, in addition to trying to overcome what I mentioned as weaknesses:

• It would probably add some value to the paper to try to simulate the mosaic of heterogeneous proliferation observed in the ferret.

• The terms complexity is used in the paper to account for the depth of the fold, I feel it is misleading in the context of the complexity of the folding pattern of the human brain.

• Clarify what is new in the model from Zarzor et al., 2021.

• The paper is using some acronyms that are annoying while reading. I suggest to stick to the actual wording: (CS = constant stiffness, HBS, etc.).

• Page 12, one paragraph has been duplicated.

• Page 14, check "to the the".

• Page 5, check "as illustrate in".

• Page 4, check "as demonstrate in".

• Page 3, the first fold appear much earlier than GW 25. See for instance Dubois, J., Benders, M., Cachia, A., Lazeyras, F., Ha-Vinh Leuchter, R., Sizonenko, S. V., … & Hüppi, P. S. (2008). Mapping the early cortical folding process in the preterm newborn brain. Cerebral cortex, 18(6), 1444-1454.

• I wonder whether your advection-diffusion model would be compatible with a Türing kind of reaction-diffusion trying to simulate the spatial heterogeneity of the proliferation with few parameters (see Lefèvre, J., & Mangin, J. F. (2010). A reaction-diffusion model of human brain development. PLoS computational biology, 6(4), e1000749.)

*Reviewer #3 (Recommendations for the authors):*

This manuscript describes a computational study of cortical folding with a particular focus on the outer subventricular zone. A novel multifield model considering cell proliferation and migration is built to simulate the cortical folding emerging process. From the model parameters analysis, the authors show that the regional difference in the thickness of the outer subventricular zone is a result of cortical folding, and the existence of the outer subventricular zone contributes to the formation of secondary mechanical instabilities. This study provides a physically significant insight into the simulation of cortical folding initiation and development, and the paper is well written. There are no major comments.

---

## [Author Response]

Essential revisions:Several assumptions underlying the model need to be discussed in more detail.1) Clarify what is new in the model from Zarzor et al., 2021.

As the title of the manuscript already indicates, compared to our previous model presented in *Zarzor et al. (2021)*, we have extended our model by including an additional proliferating zone, the outer subventricular zone (OSVZ), as a new source of cells. This particular zone in the developing human brain is currently investigated by many researchers in the field as it seems critical for the complexity of the human brain. We strongly believe that our approach, using a computational physics-based model, can help answer some of the related open questions, as now more explicitly indicated at the end of the introduction section:

“We validate our model through a comparison of the simulation results with histologically stained sections of the human fetal brain and address unresolved questions regarding the role of the OSVZ for cortical folding.”

In addition, we have implemented a new stabilization schema that allows for more accurate results and a higher sensitivity with respect to the model parameters. It further reduces the effect of the numerical spatial oscillation that appeared in Figure 9 in *Zarzor et al. (2021)*. Through the stabilization, it has become possible to generate 3D results, which we have now added to the manuscript. We have now better emphasized the numerical stabilization in the manuscript:

“After the cells reach the cortex, they diffuse isotropically, as described by the diffusion tensor ***d**_cc_* (***x***) = (*d_cc_* (*r_i_*) + *v_c_* (*c*)) ***I*** with the diffusivity *d_cc_*, the artificial viscosity *v_c_(c)*, and the second order unit tensor ***I***. The artificial viscosity term *v_c_(c)* serves as a numerical stabilization to avoid numerical oscillations associated with the advection-diffusion equation. It only acts when the actual cell density does not satisfy the balance equation and ensures more reliable results without having a particular physical meaning.”

Finally, we have significantly improved the validation approach compared to *Zarzor et al. (2021)* to enhance the relevance of our computational work for the neuroscience community.

We have now also slightly adapted the abstract and the conclusion:

“To better understand how the OSVZ affects cortical folding, we establish a multifield computational model that couples cell proliferation in different zones and migration at the cell scale with growth and cortical folding at the organ scale by combining an advection-diffusion model with the theory of finite growth. We validate our model based on data from histologically stained sections of the human fetal brain and predict three-dimensional pattern formation.”

“In this work, we have made use of a computational model for brain growth to provide insights into the role of the outer subventricular zone (OSVZ) – a unique additional proliferating zone in humans – during cortical folding in the developing brain.” and “Our simulations have systematically demonstrated how the proliferation in the OSVZ enhances the complexity of cortical folding patterns in 2D and 3D.”

2) Comment on why there is no feedback from mechanics on cell division, migration, and diffusion.

We apologize for the lack of clarity. There is indeed feedback from mechanics on the cell behavior. Firstly, the migration direction changes due to the folding of the cortex, as described in Equation 12, where n denotes the normal vector in the spatial configuration. This vector changes during the folding process and thus captures the reorientation of radial glial cell fibers, as illustrated in Figure 10 of *Zarzor et al. (2021).* In addition, the division rate in both the VZ and the OSVZ decreases due to the deformation (quantified by the maximum stretch *s* in the domain) induced by the mechanical growth problem, Equation 16.

We have now emphasized these feedback mechanisms in the Computational model section:

“The vector ***n*** represents the normalized orientation of radial glial cell fibers in the spatial configuration and controls the migration direction of neurons. As the brain grows and folds, the fiber direction changes. Through this feedback mechanism, the mechanical growth problem affects how neurons migrate and the cell density evolves locally.”

“By applying Equation 16 for the VZ, we ensure that the division rate decreases from its initial value G_vz to a smaller value as the maximum stretch value *s* in the domain increases, i.e., with increasing gestational age. This constitutes an additional feedback mechanism between the mechanical growth problem and the cell density problem: As the maximum stretch and thus the deformation increases due to constrained cortical growth, the division rate in the VZ decreases, resulting in less newborn cells” and “G^s_osvz is the division rate in the OSVZ that decreases with increasing maximum stretch *s* in the domain”.

3) Section on "How does cell proliferation in the OSVZ affect cortical folding patterns?" Clearly point out the differences between the panels in Figure 5. Add a quantification of the distance between sulci (maybe for different parameter values, you get a clearer readout). Also, comment on the mechanism underlying the decrease in distance.

Thank you for making this point. We have now edited the corresponding section and figure to include a quantification of the distance between sulci:

“In general, the distance between neighboring sulci decreases with increasing G_osvz, as marked in Figure 7. For the displayed cases, the distance decreases from d = 8.796 mm for G_osvz = 0 to d = 8.67 mm for G_osvz = 10 and finally d = 8.2 mm for G_osvz = 20. Interestingly, the cortical thickness and effective stiffness ratio at the first instability point (denoted by w in Figure 5) are the same for all these cases. Therefore, we attribute the observed differences to the faster increase in the cell density and thus cortical growth, cortical stiffness and the effective stiffness after the instability has been initiated.”

4) Please, elaborate a little on the possibility of extending the model in 3D.

This is an important point as the stabilization we have implemented indeed allows us to generate 3D results. We have now added a new figure to show that the observed trends also hold true for 3D simulations:

“Figure 8 demonstrates that the observed trends also hold true when extending the model to 3D. For the case of varying stiffness with a stiffness ratio of 3, a growth ratio of 3, and an initial division rate in the ventricular zone G_vz = 600, the folding complexity increases with increasing initial division rate in the OSVZ G_osvz.”

5) Ll. 456: "While it is to this date not clear whether this phenomenon (thickness variation of the OSVZ) is rather the cause or the result of cortical folding, our study clearly supports the latter." Since, in your model, there is no mechanism that would lead to thickness variation of the OSVZ, you can probably not make this statement. Please, clarify. However, you could state that cortical folding is sufficient to induce OSVZ thickness variations, right?

Thank you for making this valuable point. We fully agree with the reviewer and have adapted the argumentation accordingly in the Results and discussion and Conclusion sections:

“While it is to date not clear whether this phenomenon is rather the cause or the result of cortical folding, our study clearly indicates that the (mechanics-driven) process of cortical folding is sufficient to induce OSVZ variations. The forces generated by the mechanical growth problem not only fold the cortical layer but also lead to undulations in the deeper zones.”

“Consequently, our analyses suggest that the purely mechanics-driven process of cortical folding is sufficient to induce regional differences in the thickness of the OSVZ. Finally, our simulations reveal that inhomogeneous cell proliferation patterns in the OSVZ can control the location of first gyri and sulci but do not necessarily affect the distance between sulci and the overall complexity of the emerging folding pattern.”

Reviewer #1 (Recommendations for the authors):Why is there no feedback from mechanics on cell division?We apologize for the lack of clarity. The division rate in both the VZ and the OSVZ is indeed controlled by the mechanical deformation and decreases with increasing maximum stretch s, Equation 16.

We have now emphasized this mechanism in the Cell density problem section:

“By applying Equation 16 for the VZ, we ensure that the division rate decreases from its initial value G_vz to a smaller value as the maximum stretch value *s* in the domain increases, i.e., with increasing gestational age. This constitutes an additional feedback mechanism between the mechanical growth problem and the cell density problem: As the maximum stretch and thus the deformation increases due to constrained cortical growth, the division rate in the VZ decreases, resulting in less newborn cells” and “G^s_osvz is the division rate in the OSVZ that decreases with increasing maximum stretch *s* in the domain”.

P. 11: "How does cell proliferation in the OSVZ affect cortical folding patterns?" I find this section rather disappointing. Figure 5 shows hardly any difference between the folds in the presence or absence of proliferation in the OSVZ. The distance between sulci is not quantified. Maybe for different parameter values, you get a clearer readout. Also, the mechanism underlying the decrease in distance is not explained. This makes this analysis look very superficial.

Thank you for making this point. We have now edited the corresponding section and figure to include a quantification of the distance between sulci:

“In general, the distance between neighboring sulci decreases with increasing G_osvz, as marked in Figure 7. For the displayed cases, the distance decreases from d = 8.796 mm for G_osvz = 0 to d = 8.67 mm for G_osvz = 10 and finally d = 8.2 mm for G_osvz = 20. Interestingly, the cortical thickness and effective stiffness ratio at the first instability point (denoted by w in Figure 5) are the same for all these cases. Therefore, we attribute the observed differences to the faster increase in the cell density and thus cortical growth, cortical stiffness and the effective stiffness after the instability has been initiated.”

How does the presence of the OSVZ affect 3d folding patterns? Clearly, this goes beyond the work presented in this study but is the question we are really interested in. At least you should argue, why the 2d analysis is informative about the 3d case.

This is an important point as the stabilization we have implemented indeed allows us to generate 3D results. We have now added a new figure to show that the observed trends also hold true for 3D simulations:

“Figure 8 demonstrates that the observed trends also hold true when extending the model to 3D. For the case of varying stiffness with a stiffness ratio of 3, a growth ratio of 3, and an initial division rate in the ventricular zone G_vz = 600, the folding complexity increases with increasing initial division rate in the OSVZ G_osvz.”

Ll. 456: "While it is to this date not clear whether this phenomenon (thickness variation of the OSVZ) is rather the cause or the result of cortical folding, our study clearly supports the latter." Since, in your model, there is no mechanism that would lead to thickness variation of the OSVZ, you cannot make this statement, I would say. You can state that cortical folding is sufficient to induce OSVZ thickness variations.

Thank you for making this valuable point. We fully agree with the reviewer and have adapted the argumentation accordingly in the Results and discussion and Conclusion sections:

“While it is to date not clear whether this phenomenon is rather the cause or the result of cortical folding, our study clearly indicates that the (mechanics-driven) process of cortical folding is sufficient to induce OSVZ variations. The forces generated by the mechanical growth problem not only fold the cortical layer but also lead to undulations in the deeper zones.”

“Consequently, our analyses suggest that the purely mechanics-driven process of cortical folding is sufficient to induce regional differences in the thickness of the OSVZ. Finally, our simulations reveal that inhomogeneous cell proliferation patterns in the OSVZ can control the location of first gyri and sulci but do not necessarily affect the distance between sulci and the overall complexity of the emerging folding pattern.”

Reviewer #2 (Recommendations for the authors):Suggestions to authors, in addition to trying to overcome what I mentioned as weaknesses:• It would probably add some value to the paper to try to simulate the mosaic of heterogeneous proliferation observed in the ferret.

Thank you for this suggestion. We have now extended our investigations regarding heterogeneous proliferation, as highlighted above.

• The terms complexity is used in the paper to account for the depth of the fold, I feel it is misleading in the context of the complexity of the folding pattern of the human brain.

We apologize for the confusion. We did not intend to equate increasing complexity with increasing depth of the fold. To avoid misunderstandings, we have now changed the sentence to:

“In the case of constant stiffness, the effect is marginal.”

• Clarify what is new in the model from Zarzor et al., 2021.

As the title of the manuscript already indicates, compared to our previous model presented in *Zarzor et al. (2021)*, we have extended our model by including an additional proliferating zone, the outer subventricular zone (OSVZ), as a new source of cells. This particular zone in the developing human brain is currently investigated by many researchers in the field as it seems critical for the complexity of the human brain. We strongly believe that our approach, using a computational physics-based model, can help answer some of the related open questions, as now more explicitly indicated at the end of the introduction section:

“We validate our model through a comparison of the simulation results with histologically stained sections of the human fetal brain and address unresolved questions regarding the role of the OSVZ for cortical folding.”

In addition, we have implemented a new stabilization schema that allows for more accurate results and a higher sensitivity to the model parameters. It further reduces the effect of the numerical spatial oscillation that appeared in Figure 9 in *Zarzor et al. (2021)*. Through the stabilization, it has become possible to generate 3D results, which we have now added to the manuscript. We have now better emphasized the numerical stabilization in the manuscript:

“After the cells reach the cortex, they diffuse isotropically, as described by the diffusion tensor ***d**_cc_ (**x**) = (d_cc_ (r_i_) + v_c_(c)) **I*** with the diffusivity *d_cc_*, the artificial viscosity *v_c_(c)* and the second order unit tensor ***I***. The artificial viscosity term *v_c_(c)* serves as a numerical stabilization to avoid numerical oscillations associated with the advection-diffusion equation. It only acts when the actual cell density does not satisfy the balance equation and ensures more reliable results without having a particular physical meaning.”

Finally, we have significantly improved the validation approach compared to *Zarzor et al. (2021)* to enhance the relevance of our computational work for the neuroscience community.

We have now also slightly adapted the abstract and the conclusion:

“To better understand how the OSVZ affects cortical folding, we establish a multifield computational model that couples cell proliferation in different zones and migration at the cell scale with growth and cortical folding at the organ scale by combining an advection-diffusion model with the theory of finite growth. We validate our model based on data from histologically stained sections of the human fetal brain and predict three-dimensional pattern formation.”

“In this work, we have made use of a computational model for brain growth to provide insights into the role of the outer subventricular zone (OSVZ) – a unique additional proliferating zone in humans – during cortical folding in the developing brain.” and “Our simulations have systematically demonstrated how the proliferation in the OSVZ enhances the complexity of cortical folding patterns in 2D and 3D.”

• The paper is using some acronyms that are annoying while reading. I suggest to stick to the actual wording: (CS = constant stiffness, HBS, etc.).

Thank you for pointing this out. We have now stuck to the actual wording. The only acronyms we have kept are OSVZ, ORGCs, and VZ, as those are essential for our study.

• Page 12, one paragraph has been duplicated.

Thank you for this remark. We have now deleted the repeated paragraph.

• Page 14, check "to the the".

Thank you, we have corrected this mistake.

• Page 5, check "as illustrate in".

Thank you, we have corrected this mistake.

• Page 4, check "as demonstrate in".

Thank you, we have corrected this mistake.

• Page 3, the first fold appear much earlier than GW 25. See for instance Dubois, J., Benders, M., Cachia, A., Lazeyras, F., Ha-Vinh Leuchter, R., Sizonenko, S. V., … & Hüppi, P. S. (2008). Mapping the early cortical folding process in the preterm newborn brain. Cerebral cortex, 18(6), 1444-1454.

Thank you for this comment. We now have modified the sentence as follows: “the first folds appear between gestational weeks 20 and 28.”

• I wonder whether your advection-diffusion model would be compatible with a Türing kind of reaction-diffusion trying to simulate the spatial heterogeneity of the proliferation with few parameters (see Lefèvre, J., & Mangin, J. F. (2010). A reaction-diffusion model of human brain development. PLoS computational biology, 6(4), e1000749.)

This is an interesting question! The reaction-diffusion model used in the mentioned paper controls the growth factors directly. In our model, we use an advection-diffusion model for the cell-density field to explicitly describe cellular processes, i.e., cell division and migration, which only indirectly control the growth multipliers. If we adopted the mentioned model to achieve our main goal, this would make the model more complicated and would require more parameters.

In addition, if the Authors are not mistaken, during the evolution of the Turing patterns, sulci could change their location, which is not physiological. Once emerged, a sulcus will deepen but stay at the same location in the brain. Therefore, the question arises whether a Turing kind of reaction-diffusion is appropriate.